# Learning Parameterized Skills from Demonstrations

**Vedant Gupta**[1]    **Haotian Fu**[1]    **Calvin Luo**[1]    **Yiding Jiang**[2]    **George Konidaris**[1]

[1]Brown University    [2]Carnegie Mellon University

## Abstract

We present DEPS, an end-to-end algorithm for discovering parameterized skills from expert demonstrations. Our method learns parameterized skill policies jointly with a meta-policy that selects the appropriate discrete skill and continuous parameters at each timestep. Using a combination of temporal variational inference and information-theoretic regularization methods, we address the challenge of degeneracy common in latent variable models, ensuring that the learned skills are temporally extended, semantically meaningful, and adaptable. We empirically show that learning parameterized skills from multitask expert demonstrations significantly improves generalization to unseen tasks. Our method outperforms multitask as well as skill learning baselines on both LIBERO and MetaWorld benchmarks. We also demonstrate that DEPS discovers interpretable parameterized skills, such as an object grasping skill whose continuous arguments define the grasp location. [1]

## 1 Introduction

The standard application of reinforcement learning to long-horizon sequential decision-making problems often fails to leverage inherent behavioral patterns, leading to sample inefficiency. In contrast, humans exhibit a remarkable ability to extract and reuse strong priors from past experiences. For instance, training a single task-specific policy to master an Atari game can demand over 10 million samples [18, 9], whereas humans can achieve effective gameplay after merely 20 episodes. A promising avenue for explicitly learning such priors from experience lies within the options framework [27], which aims to discover modular and temporally extended skills. These skills can then be flexibly reused and composed, facilitating generalization to novel tasks and reducing the effective planning horizon.

While prior research has predominantly focused on learning either purely discrete or continuous skills, these approaches possess inherent limitations. Discrete skills alone may lack the flexibility required for broad generalization, while continuous skills can be less structured and challenging to interpret. We propose that learning *parameterized skills* [4, 5], which are discrete in nature but can be modulated by continuous arguments, offers a synergistic combination of the benefits from both independent approaches. Parameterized skills retain the structured and interpretable nature of *discrete* skills, while enabling flexible reuse in novel settings through *continuous* conditioning.

Consider the task of learning a skill to slice fruit. Generalization is crucial here, as no two fruits are identical, and the skill might need to be applied across diverse kitchen environments with varying tools. Learning a distinct discrete skill for every possible scenario is not only computationally intractable but also generalizes poorly to unseen situations. Conversely, learning a parameterized skill like `slice_fruit(x,y,z)`, where the continuous parameters specify the slicing action based on the particular instance, compactly represents a family of policies for fruit slicing. This can facilitate robust generalization to previously unseen slicing angles and accommodate different parameterizations for various fruits, tools, and kitchen conditions.

---

[1] **Website:** sites.google.com/view/parameterized-skills **Code:** github.com/guptbot/DEPS
**Correspondence:** `vedantgupta@gmail.com`

39th Conference on Neural Information Processing Systems (NeurIPS 2025).

In this work, we introduce **D**iscovery of **GE**neralizable **P**arameterized **S**kills (DEPS), an algorithm for discovering parameterized skills from expert demonstrations in an end-to-end manner. DEPS trains the three-level hierarchy illustrated in Figure 1: (i) a discrete skill selector, (ii) a continuous parameter selector conditioned on the discrete skill, and (iii) a subpolicy conditioned on both. A standard implementation of this approach is *under-specified* and often susceptible to *degenerate solutions* that minimize the behavior cloning loss without acquiring meaningful skill abstractions [11]. To mitigate these degenerate solutions, DEPS incorporates several information-theoretic constraints and architectural choices, detailed in Section 4. These include *compressing* the observation embedding before feeding it to the subpolicy networks to limit information flow and predicting continuous parameters *conditioned on each discrete skill* rather than at every timestep. These design choices compel the model to rely on the latent variables to solve the task, resulting in more robust, interpretable, and generalizable skills.

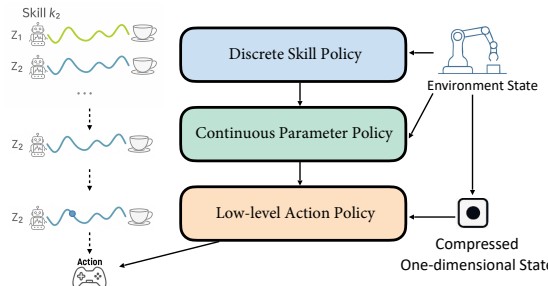

Figure 1: Three-level hierarchy of DEPS. The discrete skill policy selects a skill from the library given the full environment observation. Conditioned on that choice, the continuous-parameter policy outputs continuous parameters that modulate the chosen skill, tracing a trajectory manifold (illustrated on the left). Finally, the low-level action policy, which sees only a compressed one-dimensional robot state, produces the primitive action.

We evaluate the efficacy of DEPS in learning parameterized skills across two challenging multitask environments: LIBERO [16] and MetaWorld-v2 [30]. Our primary focus is on the rapid generalization capabilities of the learned skills, assessing their ability to adapt to novel tasks through finetuning with limited data. We demonstrate significant quantitative performance improvements over prior work in low-data regimes and provide qualitative visualizations of learned skills corresponding to fundamental actions like grasping, moving, and releasing objects. We find that DEPS consistently achieves the highest average success rate across various pretraining settings compared to existing methods, underscoring its ability to learn flexible and high-performing skills.

## 2 Related Work

A straightforward approach to learning from multitask demonstration data is to train a monolithic policy that maps a state and task label to a single action. However, previous works suggest that the difficulty of a sequential decision-making problem scales with the problem horizon [24, 10, 21]. To improve sample efficiency and generalization, many methods hence focus on learning temporally-extended action abstractions called options/skills [27].

There is a large body of work focusing on learning skills from demonstrations [14, 19, 15, 13, 26, 25, 11, 33, 6]. Notably, Shankar and Gupta [25] introduce temporal variational inference to learn skills from demonstrations. However, their method is restricted to low-dimensional state spaces. Prior work also found multi-level hierarchy to be useful for skill learning [3, 7, 1, 20, 8, 22]. All the methods above are restricted to learning a fixed number of discrete skills or continuous skills.

Early work on parameterized skills, including work by da Silva et al. [4], proposed the construction of parameterized skills by analyzing the structure of policy manifolds. However, this required labeled parameters of tasks for training. More recent methods, such as LOTUS [28] and EXTRACT [31], learn goal-conditioned discrete skills by first clustering demonstration trajectories into different discrete skills using pretrained models such as VLMs, and then learning parameterized policies corresponding to each cluster. However, this approach implicitly makes the assumption that the same discrete skill takes place in visually similar environments, which may not hold in practice. Fu et al. [5] propose to learn parameterized through online meta-learning. However, their approach requires several separate stages of training online and needs to manually design a set of tasks for each skill.

Using parameterized skills as the new action space, a large body of previous work focuses on Parameterized-action Markov Decision Process (PAMDP) [17, 2, 29, 32, 23]. These works presume a predefined library of parameterized skills, whereas our three-level hierarchy learns every policy layer end-to-end directly from raw trajectory demonstrations.

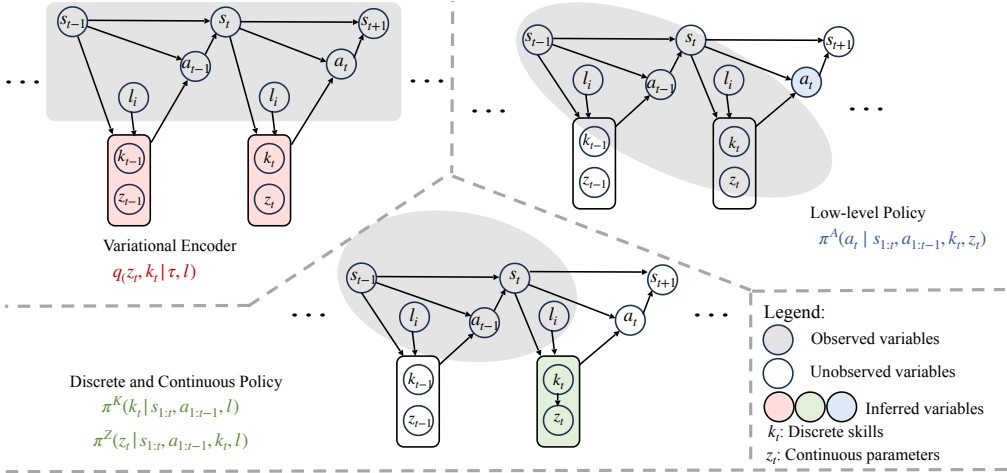

Figure 2: The underlying probabilistic graphical model of DEPS. The variational encoder has access to all the information of the trajectory from history to future. The discrete and continuous policy works as the high level policy that infers the parameterized skills based on information from previous timesteps. The low-level subpolicy infers actions based on the parameterized skills as well as the current state. Variables observable by each model are shaded in gray.

## 3 Background

We begin with deriving a training objective for learning continuously parameterized discrete skills from provided demonstration data. Consider a set of tasks $\mathcal{T}$, which are assumed to share a consistent state space, action space, and transition function. In the Learning from Demonstration setting, we assume access to a training dataset of task demonstrations $D = \{T_i\}_{i=1}^N$; each demonstration is denoted as $T_i = \{\tau_i, l_i\}$, where $\tau_i = \{s_t, a_t\}_{t=1}^{M_i}$ is the trajectory and $l_i \in \mathcal{T}$ is the task / goal description associated with the trajectory. Our goal is to learn a policy that can not only perform well for tasks covered within the training dataset $\mathcal{T}_{seen} = \bigcup \{l_i\}_{i=1}^N$, but also generalize to novel tasks $\mathcal{T}_{unseen} = \mathcal{T} \setminus \mathcal{T}_{seen}$.

For complex, long-horizon tasks $l \in \mathcal{T}$, we assume a natural decomposition into a sequence of discrete skills $\kappa = \{k_i\}_{i=1}^M$, where $k \in K$ represents a set of distinct skills. These discrete skills are further conditioned on continuous-valued parameters $z \in \mathbb{R}^d$. Our hierarchical approach comprises three main components:

1. A **discrete policy** $\pi^K(k_t \mid s_{1:t}, a_{1:t-1}, l)$ that predicts the discrete skill.

2. A **continuous policy** $\pi^Z(z_t \mid s_{1:t}, a_{1:t-1}, k_t, l)$ that predicts the continuous parameters for the chosen discrete skill.

3. A **subpolicy** $\pi^A(a_t \mid s_{1:t}, a_{1:t-1}, k_t, z_t)$ that generates an executable action based on the current discrete skill and its continuous parameterization.

During interaction, the agent first samples a discrete skill $k_t \sim \pi^K$, then a continuous parameter $z_t \sim \pi^Z$, and finally an action $a_t \sim \pi^A$.

## 4 Discovery of GEneralizable Parameterized Skills (DEPS)

We introduce DEPS, a framework for learning continuously parameterized skills from demonstrations. DEPS jointly trains a hierarchical policy by maximizing a variational lower bound on the likelihood of demonstrated trajectories. This section details the derivation of our training objective using temporal variational inference and then describes our approach to modeling skills as parameterized trajectory manifolds, a key component for generalization.

### 4.1 Variational Inference for Parameterized Skill Discovery

Consider a sampled demonstration $T = \{\tau, l\} \in D$, where $\tau = \{s_t, a_t\}_{t=1}^M$, and some corresponding sequence of discrete skills $\kappa = \{k_i\}_{i=1}^M$ and continuous parameters $\zeta = \{z_i\}_{i=1}^M$. Then, we can write the joint likelihood of these sequences $p(\tau, \kappa, \zeta, l)$ as follows:

$$p(\tau, \kappa, \zeta, l) = p(s_1, l) \prod_{t=1}^M \{\pi^K(k_t|s_{1:t}, a_{1:t-1}, k_{1:t-1}, z_{1:t-1}, l)\pi^Z(z_t|s_{1:t}, a_{1:t-1}, k_{1:t}, z_{1:t-1}, l)$$
$$\pi^A(a_t|s_{1:t}, a_{1:t-1}, k_t, z_t)p(s_{t+1}|s_t, a_t)\}. \tag{1}$$

To train policies $\pi^K, \pi^Z$, and $\pi^A$ that are autoregressively usable at inference time, each policy can only be conditioned on prior observations, actions, discrete skills, and continuous parameters. In line with prior literature, we aim to maximize the objective $\mathbb{E}_{(\tau, l) \sim D}\left[\log p(\tau, l)\right]$. However, calculating $\log p(\tau, l)$ exactly involves an intractable marginalization over all possible sequences of discrete and continuous skills. We use temporal variational inference [25] to estimate $\mathbb{E}_{(\tau, l) \sim D}\left[\log p(\tau, l)\right]$ while maintaining the autoregressive structure of the policies learned. We introduce a *variational distribution* $q(\kappa, \zeta|\tau, l)$ which is meant to approximate $p(\kappa, \zeta|\tau, l)$. Due to the non-negativity of KL divergence:

$$\mathbb{E}_{(\tau, l) \sim D}\left[\log p(\tau, l)\right] \geq \mathbb{E}_{(\tau, l) \sim D}\left[\log p(\tau, l) - D_{KL}(q(\kappa, \zeta|\tau, l)||p(\kappa, \zeta \mid \tau, l))\right] := \mathcal{L}.$$

Note that the bound above is tight when $q(\kappa, \zeta|\tau, l) = p(\kappa, \zeta|\tau, l)$. Now,

$$\mathcal{L} = \mathbb{E}_{(\tau, l) \sim D,\, (\kappa, \zeta) \sim q(\kappa, \zeta|\tau, l)}\left[\log p(s_1, l) + \sum_{t=1}^M \left\{\log \pi^K\left(k_t \mid \mathcal{H}_t^K\right) + \log \pi^Z\left(z_t \mid \mathcal{H}_t^Z\right)\right.\right.$$
$$\left.\left. + \log \pi^A\left(a_t \mid \mathcal{H}_t^A\right) + \log p(s_{t+1} \mid s_t, a_t)\right\} - \log q(\kappa, \zeta \mid \tau, l)\right]. \tag{2}$$

where $\mathcal{H}_t^K = (s_{1:t}, a_{1:t-1}, k_{1:t-1}, z_{1:t-1}, l)$, $\mathcal{H}_t^Z = (s_{1:t}, a_{1:t}, k_{1:t}, z_{1:t-1}, l)$, and $\mathcal{H}_t^A = (s_{1:t}, a_{1:t-1}, k_t, z_t)$, and where the last step uses the joint distribution derived in Equation 1.

Keeping computational efficiency during training in mind, to enable the discrete skills and continuous parameters for each timestep to be sampled in parallel, we assume that conditional on previous states and actions, $(k_t, z_t)$ is independent of $(k_i, z_i)$ for $i < t$. This allows us to rewrite $\pi^K(k_t|s_{1:t}, a_{1:t-1}, k_{1:t-1}, z_{1:t-1}, l)$ as $\pi^K(k_t|s_{1:t}, a_{1:t-1}, l)$ and $\pi^Z(z_t|s_{1:t}, a_{1:t-1}, k_{1:t}, z_{1:t-1}, l)$ as $\pi^Z(z_t|s_{1:t}, a_{1:t-1}, k_t, l)$. Since the subpolicy $\pi^A$ is only conditioned on the *current* continuous and discrete parameters, one can show that for any fixed $\pi^A$, the joint likelihood in Equation 1 can be maximized by the simplified expressions for $\pi^K$ and $\pi^Z$ above. Our empirical results in Section 5 also validate that this assumption does not prevent useful skills from being learned.

Since $q(\kappa, \zeta|\tau, l)$ is meant to approximate $p(\kappa, \zeta|\tau)$, one can now rewrite $q(\kappa, \zeta|\tau, l)$ as follows:

$$q(\kappa, \zeta|\tau, l) = \prod_{t=1}^M q(k_t|\tau, l)q(z_t|\tau, k_t, l). \tag{3}$$

Using Equation 3 and the observation that the dynamics $p(s_{t+1}|s_t, a_t)$ and the joint distribution of the task and initial state, $p(s_1, l)$, do not affect the gradient of our loss, we can rewrite Equation 2 in the following, cleaner, form:

$$\mathcal{L} = \mathbb{E}_{(\tau, l) \sim D,\, (\kappa, \zeta) \sim q(\kappa, \zeta|\tau, l)}\left[\sum_{t=1}^M \log \pi^A(a_t \mid \mathcal{H'}_t^A)\right] - \mathbb{E}_{(\tau, l) \sim D}\left[\sum_{t=1}^M D_{\mathrm{KL}}\left(q(k_t \mid \tau, l) \parallel \pi^K(\cdot \mid \mathcal{H'}_t^K)\right)\right.$$
$$\left. - \mathbb{E}_{k_t \sim q(k_t|\tau, l)}\left[D_{\mathrm{KL}}\left(q(z_t \mid \tau, k_t, l) \parallel \pi^Z(\cdot \mid \mathcal{H'}_t^Z)\right)\right]\right], \tag{4}$$

where $\mathcal{H'}_t^A = (s_{1:t}, a_{1:t-1}, k_t, z_t), \mathcal{H'}_t^K = (s_{1:t}, a_{1:t-1}, l), \mathcal{H'}_t^Z = (s_{1:t}, a_{1:t-1}, k_t, l)$.

We represent the discrete skills using categorical distributions and the continuous parameters using Gaussian distributions. Hence, the KL-divergence terms above can be calculated exactly. We therefore minimize this loss to learn the discrete and continuous policies as well as the low-level policy.

## 4.2 Skills as Parameterized Trajectory Manifolds

A common failure mode in learning skills from demonstrations is for the low-level policy to minimize behavior cloning loss without discovering meaningful, generalizable skill abstractions. This often occurs when state spaces for different tasks have minimal overlap, allowing a high-capacity policy to memorize task-specific behaviors in different state space subsets. To address this, DEPS introduces an information bottleneck in the subpolicy and models skills as manifolds of parameterized trajectories.

**Information Bottleneck via State Compression.** Our core strategy is to provide the subpolicy $\pi^A(a_t | s'_t, k_t, z_t)$ with a *compressed*, lower-dimensional version of the current state, $s'_t$, instead of the full observation $s_t$. This aims to:

1. **Enhance State-Space Overlap and Generalization:** Mapping raw states into a shared, compressed space increases input distribution overlap across tasks, resulting in policies that generalize to novel tasks within or near this compressed manifold.

2. **Force Reliance on Latent Variables:** A highly compressed state $s'_t$ alone is often insufficient to determine the correct action $a_t$. The subpolicy must rely on the discrete skill $k_t$ and continuous parameter $z_t$ to resolve ambiguity, compelling these latents to encode crucial skill-related information.

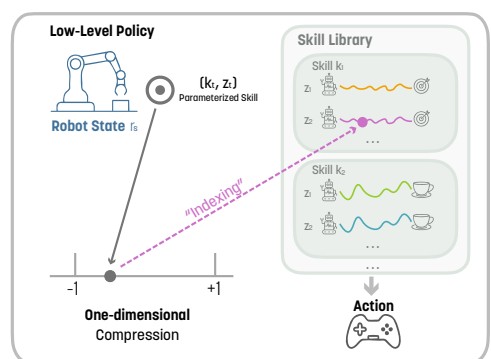

Figure 3: Skills as Parameterized Trajectory Manifolds. We hypothesize that a single skill corresponds to a family of parameterized trajectories. A one-dimensional state representation indexes into this generalizable manifold to predict actions.

**Conceptualizing Skills.** What distinguishes a versatile "skill" from a collection of disparate behaviors? We posit that a versatile skill exhibits a lower-dimensional structure. *Trajectories* from executing a coherent skill (e.g., "pick up a toy" from various positions) likely lie on or near a common low-dimensional manifold, with variations corresponding to different paths or modulations. In contrast, trajectories from disparate behaviors (e.g., "clean dishes" and "pick up toys") belong to different manifolds. A parameterized skill $(k, z)$ thus selects and refines a family of trajectories sharing a common structure. Movement along such a trajectory can be indexed by a low-dimensional variable (e.g., progress), motivating aggressive state compression.

**Projective State Compression to One Dimension.** Given the discrete skill $k_t$, continuous parameter $z_t$, and a feature vector $s_t^{\text{proj}}$ from the raw observation $s_t$ (e.g., robot end-effector coordinates), we compress $s_t^{\text{proj}}$ into a scalar $s'_t$:

$$s'_t = \tanh\left(\mathbf{w}_{(k_t, z_t)} \cdot s_t^{\text{proj}} + b_{(k_t, z_t)}\right). \tag{5}$$

Here, $\mathbf{w}_{(k_t, z_t)}$ (a vector) and $b_{(k_t, z_t)}$ (a scalar bias) are outputs of a small MLP, $f_{\text{compress}}(k_t, z_t)$. The vector $\mathbf{w}_{(k_t, z_t)}$ defines a skill-specific projection axis. The $\tanh$ activation normalizes $s'_t$ to $[-1, 1]$, ensuring a bounded index. This 1D compressed state $s'_t$, along with $k_t$ and $z_t$, is fed to a feedforward network to output action distribution parameters. This forces $k_t$ and $z_t$ to encode task-relevant information not captured by the highly compressed $s'_t$.

## 4.3 Overall Framework and Flow

The DEPS framework consists of four interconnected neural network components: a variational network ($q$), a discrete policy network ($\pi^K$), a continuous policy network ($\pi^Z$), and a subpolicy network ($\pi^A$). These are trained jointly using the objective in Equation 4.

The **variational network** $q(\kappa, \zeta | \tau, l)$ is implemented as a bidirectional GRU. It processes entire demonstration trajectories to infer posterior distributions over skills $k_t$ and parameters $z_t$. To encourage temporally extended skills, continuous parameters $z_t$ are predicted per discrete skill $k$ rather than per timestep within a trajectory. This prevents the continuous parameters from rapidly changing, and potentially embedding the action to take at each timestep. Additionally, to prevent the continuous parameters from overfitting to specific trajectories, we introduce a *Skill Parameter Norm Penalty* to discourage large-magnitude continuous parameters.

The **discrete policy** $\pi^K(k_t|s_{1:t}, a_{1:t-1}, l)$ and **continuous policy** $\pi^Z(z_t|s_{1:t}, a_{1:t-1}, k_t, l)$ form the high-level policy. In practice, the entire conditioning context is often unnecessary; in our experiments, we find that removing the history of actions from $\pi^K$ and $\pi^Z$ still produces high empirical performance while simplifying the implementation. Both $\pi^K$ and $\pi^Z$ are implemented as unidirectional GRUs that process the history of states. Their outputs, combined with the current task $l$ (and $k_t$ for $\pi^Z$), are passed through MLPs to predict the current skill $k_t$ and its parameters $z_t$ autoregressively. The **low-level subpolicy** $\pi^A(a_t|s'_t, k_t, z_t)$ executes the chosen skill.

Unlike the variational network's strategy of predicting one set of continuous parameters per skill instance for an entire trajectory, the continuous policy network $\pi^Z$ predicts continuous parameters *for every timestep*. This allows for refinement of the continuous parameter $z_t$ at each step based on the most recent observation, enabling more reactive behavior during inference.

Crucially, DEPS incorporates an **information asymmetry**: the high-level policies (variational network, discrete policy, and continuous policy) have access to both rich image observations if available and the robot's proprioceptive state. In contrast, the low-level subpolicy $\pi^A$ only observes the robot's proprioceptive state (which is then compressed into $s'_t$). This restriction forces the subpolicy to rely on $(k_t, z_t)$ for skill-specific guidance and prevents it from overfitting to visual details that might hinder generalization. By learning to operate from a more abstract, compressed state representation conditioned by the latent skill variables, the subpolicy is encouraged to learn more generalizable behaviors. Appendix A contains additional details, along with the pretraining and finetuning procedures.

## 5 Experiments

We empirically evaluate the ability of DEPS to generalize to novel tasks with minimal finetuning across two challenging multitask environments, LIBERO [16] and MetaWorld-v2 [30]. Within each benchmark, we first pretrain on a series of tasks to learn parameterized skills. The pretrained model is then finetuned on an unseen task for 500 gradient steps, and its performance is evaluated with 20 rollouts after every 50 steps of finetuning (for a total of 10 evaluations). Each rollout is assigned a binary reward based on whether the task was successfully completed. We evaluate the performance of our approach on a series of novel and previously used evaluation sets. For each set of evaluation tasks, we report two metrics:

- **Mean Success**: The average success rates across all unseen tasks, averaged across all 10 evaluations. This measures the algorithm's consistency.
- **Mean Highest Success**: The highest success rate across all 10 evaluations of a given task, averaged across all unseen tasks. This measures the performance of the algorithm when using the optimal number of gradient steps during finetuning for each specific task. Previous work [16, 33] uses this metric to evaluate performance.

We report both metrics along with their standard deviation across evaluated seeds. We find that DEPS displays consistently superior rapid generalization to unseen tasks across diverse evaluation settings.

### 5.1 Baselines

We compare our approach against the following baselines (1) A multitask behavior cloning network (`BC`), (2) the same multitask BC architecture, but *without any pretraining* (`BC-Untrained`), and (3) PRISE [33], the state-of-the-art baseline that learns action "tokens" and then applies byte pair encoding to learn common action sequences (`PRISE`). Information on the specific hyperparameters used and steps taken to ensure a fair comparison across baselines can be found in Appendix C.

### 5.2 Experiment Setting

**LIBERO** LIBERO is a multitask benchmark involving various object manipulation tasks with a robot arm (visualizations of example tasks are provided in Figure 4) across visually diverse environments. To pre-train model architectures, we use 80 tasks from LIBERO-90 using the offline dataset provided by Liu et al. [16]. For each of the 80 pretraining tasks, the dataset provides 50 demonstrations collected using human tele-operation. The state-space for each task includes images from two different viewpoints, a 9-dimensional vector representing the robot's state, and a language description of the task at hand. The action space consists of a 6-dimensional real-valued vector representing desired arm movements, along with a binary variable to open/close the gripper. We perform 20 passes over the pretraining data for each method.

Table 1: Average success rate across evaluation settings on LIBERO and MetaWorld-v2. All results are averaged across 5 seeds.

| Evaluation Set | Algorithm | Mean Success | Mean Highest Success |
|---|---|---|---|
| LIBERO-OOD | DEPS | **0.34** ± 0.08 | **0.66** ± 0.12 |
| | PRISE | 0.10 ± 0.09 | 0.27 ± 0.23 |
| | BC | 0.15 ± 0.04 | 0.36 ± 0.08 |
| | BC-Untrained | 0.01 ± 0.00 | 0.08 ± 0.02 |
| LIBERO-10 | DEPS | **0.08** ± 0.04 | **0.24** ± 0.09 |
| | PRISE | 0.02 ± 0.02 | 0.07 ± 0.06 |
| | BC | 0.07 ± 0.02 | 0.18 ± 0.03 |
| | BC-Untrained | 0.00 ± 0.00 | 0.01 ± 0.00 |
| LIBERO-3-shot | DEPS | **0.26** ± 0.03 | **0.49** ± 0.03 |
| | PRISE | 0.07 ± 0.07 | 0.19 ± 0.14 |
| | BC | 0.11 ± 0.05 | 0.22 ± 0.08 |
| | BC-Untrained | 0.01 ± 0.00 | 0.03 ± 0.01 |
| MW-Vanilla | DEPS | **0.45** ± 0.03 | **0.65** ± 0.03 |
| | PRISE | 0.21 ± 0.07 | 0.33 ± 0.10 |
| | BC | 0.35 ± 0.02 | 0.51 ± 0.01 |
| | BC-Untrained | 0.25 ± 0.02 | 0.45 ± 0.05 |
| MW-PRISE | DEPS | **0.32** ± 0.03 | **0.53** ± 0.03 |
| | PRISE | 0.06 ± 0.02 | 0.17 ± 0.05 |
| | BC | 0.25 ± 0.02 | 0.41 ± 0.03 |
| | BC-Untrained | 0.12 ± 0.01 | 0.29 ± 0.01 |

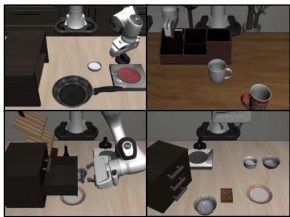

Figure 4: Images of example tasks from LIBERO

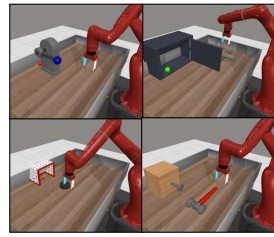

Figure 5: Images of example tasks from MetaWorld-v2

Pretrained checkpoints are then evaluated on the following three test settings:

- **LIBERO-OOD:** 10 unseen tasks from LIBERO-90 involving previously unseen environments and objects, making it a strong test of generalization to out-of-distribution tasks. Descriptions of the tasks in this set can be found in Appendix D. Each task comes with 50 expert demonstrations.
- **LIBERO-10:** The standard LIBERO evaluation dataset, consisting of long-horizon tasks, which are mostly concatenations of tasks seen in the pretraining set, measuring the ability of an algorithm to accurately complete *in-distribution* but long-horizon tasks. Each task comes with 50 expert demonstrations
- **LIBERO-3-shot**: This dataset consists of the tasks in LIBERO-OOD but with only 3 demonstrations per task, testing the ability to successfully learn new tasks with minimal data.

**MetaWorld-v2**  MetaWorld-v2 is a multitask benchmark involving various object manipulation tasks with a robot arm (visualizations of example tasks are provided in Figure 5). To evaluate performance on MetaWorld-v2, we utilize the provided expert scripted policies provided in MetaWorld, collecting 50 demonstration trajectories for each task. The state-space for each task includes an image and an 8-dimensional vector representing the robot's state, while the action space is a 4-dimensional real-valued vector. We pretrain each method on a set of 10 tasks, performing 40 passes over the training data. We then evaluate the performance of pretrained checkpoints on two different evaluation sets described below (information on the specific tasks used for pretraining and finetuning can be found in Appendix D). Due to the shorter average time horizon and lower task diversity compared to LIBERO, we focus on the 3-shot fine-tuning performance in both evaluation sets.

- **MW-Vanilla**: A standard set of 5 unseen tasks proposed in MetaWorld.
- **MW-PRISE**: 5 unseen tasks as used in PRISE, consisting of longer horizon tasks on average.

### 5.3  Primary Experimental Results

Table 1 shows the mean success and mean highest success rate with minimal finetuning on each of the evaluation datasets outlined in Section 5.2. DEPS achieves a *significantly* higher mean success and mean highest success rate across evaluation regimes, showing the effectiveness of parameterized skills and observation-space compression in learning generalizable abstractions. Notably, in LIBERO-OOD, which tests rapid generalization to out-of-distribution environments and objects, DEPS achieves a mean success rate of 0.34, which is more than double that of the standard BC approach (0.15) and more than triple the performance of PRISE (0.10). Even in the extremely data-scarce LIBERO-3-shot setting, DEPS maintains robust performance (0.26 mean success), outperforming both PRISE (0.07) and standard BC (0.11) by substantial margins. DEPS maintains its advantage across both evaluation sets in MetaWorld-v2, where tasks have a generally shorter horizon.

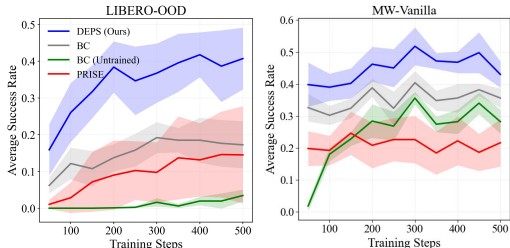

Figure 6: Average success rates as a function of the number of finetuning gradient steps taken for LIBERO-OOD (left) and MW-Vanilla (right).

Table 2: Robustness of parameterized skills over different total pretraining epochs (PT). Results use 50-shot finetuning over LIBERO-OOD and are averaged across 3 seeds.

| PT | Algorithm | Mean Success | Mean Highest Success |
|----|-----------|--------------|----------------------|
| 5  | DEPS  | $0.24 \pm 0.08$ | $0.64 \pm 0.09$ |
|    | PRISE | $0.02$ [upto $0.05$] | $0.11 \pm 0.13$ |
|    | BC    | $0.07 \pm 0.03$ | $0.30 \pm 0.08$ |
| 10 | DEPS  | $0.39 \pm 0.02$ | $0.75 \pm 0.01$ |
|    | PRISE | $0.12 \pm 0.15$ | $0.27 \pm 0.33$ |
|    | BC    | $0.10 \pm 0.06$ | $0.30 \pm 0.12$ |
| 15 | DEPS  | $0.39 \pm 0.05$ | $0.74 \pm 0.04$ |
|    | PRISE | $0.12 \pm 0.14$ | $0.33 \pm 0.26$ |
|    | BC    | $0.12 \pm 0.06$ | $0.32 \pm 0.07$ |

The relatively poor performance of PRISE, despite its strong results in Zheng et al. [33], highlights the challenges of rapid generalization compared to the more extensive fine-tuning procedures used in prior work. Similarly, the near-zero performance of the untrained BC baseline on LIBERO tasks (0.01 mean success on LIBERO-OOD) underscores the importance of pretraining for successful adaptation in complex environments.

Figure 6 shows the mean success rates on LIBERO-OOD and MW-Vanilla as a function of the number of fine-tuning gradient steps taken (averaged across 5 seeds). DEPS outperforms other baselines *irrespective* of the number of gradient steps taken. These results collectively demonstrate that DEPS provides a robust foundation for rapid generalization across diverse robotic manipulation tasks.

### 5.4 Robustness to Pretraining Amounts

While the results in Section 5.3 use 20 epochs of pretraining for LIBERO and 40 epochs of pretraining for MetaWorld-v2, we also evaluate the performance of our approach on smaller pretraining amounts to evaluate its data efficiency and robustness to different pretraining amounts. Table 2 shows the performance of DEPS and evaluated baselines on smaller pretraining amounts. DEPS consistently outperforms evaluated baselines across pretraining amounts. This same observation also holds across MetaWorld (results can be found in Appendix E). Notably, with limited pretraining (e.g. 5 epochs), the margin between our method and evaluated baselines *increases*. This suggests that, in addition to improving generalization to unseen tasks, learning parameterized skills with compression might also increase the data efficiency of pretraining.

### 5.5 Additional Quantitative Results

We provide additional experimental results in the Appendix that suggest that (i) compression to 1D state is essential to DEPS' performance (Appendix F), (ii) changing the limit on the maximum number of discrete skills considerably improves the performance of DEPS, suggesting improvements over the presented results may occur with a hyperparameter sweep (Appendix G), (iii) learning only discrete or only continuous skills does *not* replicate DEPS' performance (Appendix H), and DEPS maintains its performance advantage on increasing the number of finetuning gradient steps (Appendix I).

### 5.6 Qualitative analysis of Learned Parameterized Skills

In addition to its strong quantitative performance, DEPS also learns interpretable parameterized skills, with discrete skills and continuous parameters encoding *skill-relevant* information. To this end, we provide visualizations showing the (i) segmentation of tasks into intuitive discrete skills, (ii) smooth variations in the skill policy on varying the continuous parameter (iii) high overlap in the continuous parameters used across tasks and (iv) monotonicity in the learned compressed state embeddings.

**Analysis of Learned Trajectory Segmentations.** We find that DEPS discovers intuitive parameterized skills. In LIBERO, the learned skills correspond to primitive behaviors such as grasping, moving, and releasing objects. The same discrete skills are used consistently across environments and object types, with changing continuous parameters to encode task-specific details. We provide a representative visualization of the decomposition of trajectories into skills in Figure 7, and more visualizations can be found on our project website (sites.google.com/view/parameterized-skills).

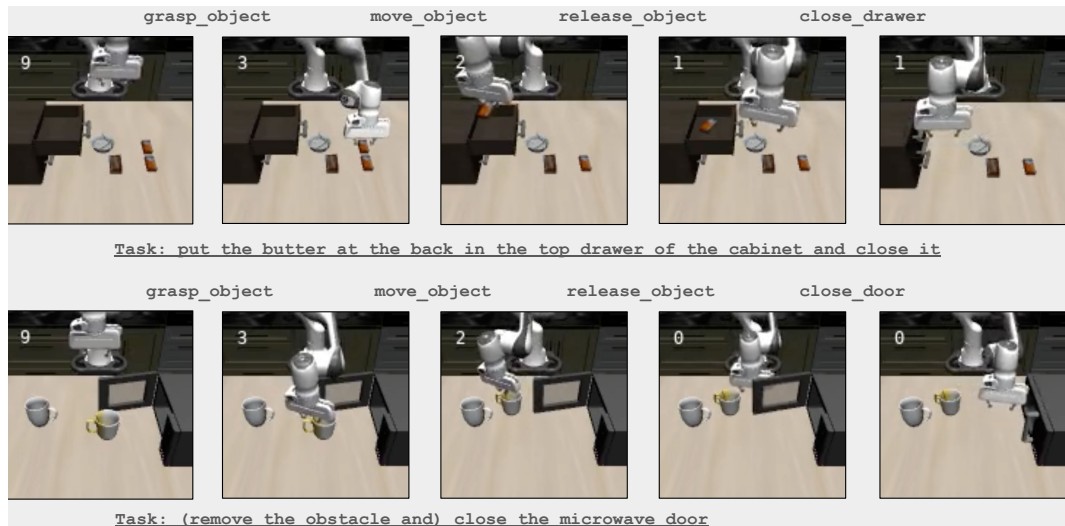

Figure 7: Visualization of the discovered discrete skills for two sample tasks. Images represent the points at which the sampled discrete skill changes, with the discrete skill taking place between two images indicated on top (DEPS labels discrete skills as integers, which are visible in the figure. The natural language skills were selected to be consistent with the discovered segmentations of the demonstration trajectories). We find that general skills corresponding to grasping, moving, and releasing objects are learned and robustly applied across visually diverse tasks. Furthermore, for less frequent subtasks such as closing drawers and microwaves, independent discrete skills are discovered.

**Analysis of Learned Continuous Parameterizations** We find that changing the continuous parameters provided to a single discrete skill results in smooth variations in the resulting policy. For example, changing the continuous parameters to the discovered skill to grasp objects smoothly varies the resulting grasp location. Additionally, for a given skill, we find that there is overlap in the continuous parameters used for different tasks, suggesting that the continuous parameters encode *skill-relevant* information, not *task* or *environment relevant* information. Visualizations of this can be found on our project website.

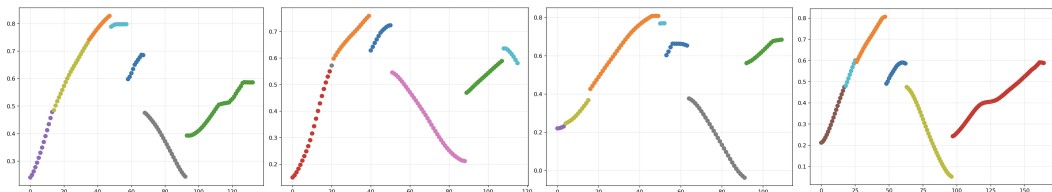

Figure 8: Plot of the learned 1-dimensional compressed state indices (y-axis) with respect to the time-step (x-axis). Each plot represents a single trajectory from a different task in LIBERO-90, and each colour corresponds to a different discrete skill. We find that the compressed state changes monotonically within a discrete skill, suggesting its use as an index.

**Analysis of Learned Observation Compressions** We verify whether the learned 1D compressed states function as "indices" into a trajectory as outlined in Section 4.2. Figure 8 shows the learned compressed state for single trajectories as a function of the timestep. We find that the state embeddings for a single skill monotonically increase/decrease at roughly uniform rates as the timestep increases, as would be expected from an index into a trajectory. Furthermore, we find sudden changes in the learned compressed state when the selected discrete skill and continuous parameter switches, which is consistent with our expectation that the index should "reset" on changing the trajectory represented by the current parameterized skill.

In Appendix F, we provide an ablation of the performance of DEPS with and without compression to one-dimensional state. We find that our compression method significantly improves the task-generalization ability of the learned parameterized skills.

# 6 Conclusion

We present DEPS, an end-to-end method to learn parameterized skills from demonstration data. To do this, we initially derive a loss function using temporal variational inference; however, simultaneously, we note that directly utilizing this loss function admits *degenerate solutions*. To avoid this, we utilize a series of information-theoretic methods. Notably, we present a novel view of skills as families of similar trajectories, motivating the aggressive compression of observations to one-dimensional "indices." We evaluate the resulting algorithm across a diverse and rigorous set of evaluation using LIBERO and MetaWorld-v2, measuring its performance on unseen tasks under (i) different pre-training/finetuning task splits, (ii) different pretraining budgets (iii) different finetuning budgets (iv) different amounts of finetuning data and (v) different measures of success (Mean Success and Mean Highest Success). We find that DEPS consistently outperforms existing baselines across evaluation settings. The performance improvement offered by DEPS is especially notable when evaluated on out-of-distribution tasks (LIBERO-OOD, Table 1), low-data regimes (LIBERO-3-shot, Table 1), and limited compute availability (Table 2). We discuss limitations of our approach in Appendix J. We believe our approach shows promise towards better data-efficient task generalization in robotics.

## Acknowledgments and Disclosure of Funding

This project was supported by ONR REPRISM MURI N00014-24-1-2603 and ONR grant N00014-22-1-2592. YJ was supported by the Google PhD fellowship. This work was conducted using computational resources and services at the Center for Computation and Visualization, Brown University.

We thank the anonymous reviewers of this work for their helpful comments, which helped us strengthen and clarify this work, We would also like to thank the authors of LIBERO and MetaWorld for developing and open-sourcing the benchmarks used in our experiments.

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

# A Implementation Details

## A.1 Variational Network

The variational network learns the distribution $q(\kappa, \zeta | \tau, l)$. We implement this as a bidirectional GRU that takes as input the encoded states and actions corresponding to a sampled demonstration trajectory $\tau$ as well as a task ID (or the provided task embedding in the case of LIBERO), denoted by $l$. At each timestep $t$, the outputs of the GRU are passed through a prediction head to predict a categorical distribution of the discrete skill at that particular timestep $t$. A separate output head is used to predict the continuous parameters. However, to prevent the continuous parameters from changing at each timestep, we predict the continuous parameters *for each discrete skill instead of each timestep*. This "restricts" the amount of information conveyable through the discrete skills and continuous parameters that encode a trajectory, forcing the subpolicy networks to learn *temporally extended* policies. [2] We model the continuous parameters as Gaussian distributions using the reparameterization trick [12].

Once the discrete skills are sampled, they can be used as indices into the set of predicted continuous parameters to choose the associated continuous parameter for each timestep. [3]

As training progresses, we observe that the average distance between continuous parameters increases, possibly indicating overfitting. To encourage compact and generalizable representations, we introduce a *Skill Parameter Norm Penalty* that discourages large-magnitude continuous parameters. Formally, we add the following regularization term to the loss in Equation 4: $\mathcal{L}_{\text{norm}} = \lambda_{\text{norm}} \sum_{t=1}^{M} \|z_t\|^2$, where $z_t$ denotes the continuous parameters active at timestep $t$ (corresponding to the skill $k_t$) as predicted by the variational network for that skill instance, and $\lambda_{\text{norm}}$ is a regularization coefficient.

## A.2 Discrete and Continuous Networks

DEPS implements the skill distribution $\pi^K(k_t | s_{1:t}, a_{1:t-1}, l)$ and parameterization distribution $\pi^Z(z_t | s_{1:t}, a_{1:t-1}, k_t, l)$ with a discrete and continuous neural network, respectively. Both models have the same architecture, implemented as a one-directional GRU, which takes in as input the history of states. The output of the GRU at each timestep is then concatenated with the task ID $l$ and passed through an MLP; for the discrete skill network, this MLP predicts a categorical distribution with $|K|$ possible values, from which $k_t$ is sampled at each timestep. For the continuous skill network this MLP predicts a set of parameters for each of the $|K|$ skills. In practice, we find that it is sufficient to only predict the mean for each dimension of the continuous parameters, i.e. $|K| \times d$ values for a $d$-dimensional continuous parameterization. We then index into the parameters corresponding to the sampled discrete skill $k_t$ at each timestep.

Unlike the variational network's strategy of predicting one set of continuous parameters per skill instance for an entire trajectory, the continuous policy network $\pi^Z$ predicts continuous parameters *for every timestep*. This allows for refinement of the continuous parameter $z_t$ at each step based on the most recent observation, enabling more reactive behavior during inference.

## A.3 Low-Level Policy Details

The subpolicy network $\pi_\theta^A$ learns the distribution $\pi^A(a_t | s_t', k_t, z_t)$.. However, we find that empirical performance improves by removing $k_t$ as a direct input to the subpolicy network. Including $k_t$ can lead to optimization pitfalls such as collapse to a single discrete skill or training instability.

We note that explicitly removing $k_t$ from the set of conditioning variables does not reduce the information available to $\pi_\theta^A$, as the continuous parameters corresponding to different discrete skills occupy disjoint subspaces in practice. Under this formulation, the discrete skill acts as a "selector" that indexes into one of $|K|$ subspaces of continuous parameters, where each subspace represents

---

[2] While this restriction of one continuous parameter value per discrete skill per trajectory works in our experiments, it is possible that longer-horizon trajectories will require the same discrete skill to be used with multiple continuous parameterizations. However, the same principle easily extends to this setting, as one can predict $n$ sets of continuous parameters per discrete skill per trajectory, where $n$ is a chosen hyperparameter.

[3] Another key difference between our architecture and that used by Shankar and Gupta [25] is that we do not predict binary variables representing skill termination at each timestep. We find that including this variable makes it significantly harder to get training to stabilize by adding dependencies between different time steps, which also slows the implementation down.

a fundamentally different skill type. Conditioning on the continuous parameter alone can thus still expose information about the disjoint space it inhabits and therefore the associated discrete skill being utilized.

Hence, $\pi_\theta^A$ takes the 1D compressed state $s_t'$ and the continuous parameter $z_t$. This combined vector is processed by a feedforward neural network (the "Markov policy network" head) that outputs the parameters of the action distribution (e.g., mean and variance for a Gaussian action space, or mixture parameters for a Gaussian Mixture Model).

### A.4 Pretraining (Skill Discovery)

Pretraining involves optimizing the objective in Equation 4 on the provided expert demonstration dataset $D$. During this phase, the sequences of discrete skills $\kappa$ and continuous parameters $\zeta$ are sampled from the *variational network* $q(\kappa, \zeta | \tau, l)$. The bidirectional nature of $q$, which considers the entire trajectory (both past and future), is crucial for discovering meaningful skill abstractions. The first term in Equation 4 (the reconstruction term for actions) encourages the subpolicy $\pi^A$ and the variational posteriors $q$ to jointly explain the actions in the demonstrations. The KL-divergence terms ensure that the autoregressive policies $(\pi^K, \pi^Z)$ learn to predict skill sequences similar to those inferred by $q$, while also regularizing $q$ towards distributions that can be effectively modeled by these autoregressive policies at inference time.

### A.5 Finetuning (Task Generalization)

For generalizing to new, potentially unseen tasks, we finetune the learned policies $(\pi^K, \pi^Z, \pi^A)$. For task id $l$, in MetaWorld we use a one-hot vector for all the 50 tasks, while in LIBERO we use the provided latent language embeddings as done in previous work. The finetuning process also optimizes the objective in Equation 4. However, a key difference from pretraining is that the discrete skills $\kappa$ and continuous parameters $\zeta$ are now sampled from the *autoregressive policies* $\pi^K(k_t | s_{1:t}, a_{1:t-1}, l)$ and $\pi^Z(z_t | s_{1:t}, a_{1:t-1}, k_t, l)$, respectively, instead of from the variational network $q$. This aligns the training conditions during finetuning more closely with the actual inference procedure, where $q$ is not used. While sampling from $\pi^K$ and $\pi^Z$ for training the subpolicy can be biased, we empirically find that this strategy improves performance on unseen tasks at inference time.

## B    Task Prediction from Observations

To motivate the need to provide a compressed version of the observation space to subpolicy networks, we study the overlap in the observation space across different tasks in LIBERO-90. Specifically, for each task in LIBERO-90, we split the 50 provided demonstration trajectories into 45 training trajectories and 5 test trajectories. Using the $45 * 90 = 4050$ trajectories in the training set, we train a simple classifier to predict which of the 90 tasks a **single observation** belongs to. Our model consists of the same Resnet-18 image encoder used in our experiments, with a two-layer MLP on top (with a hidden size of 1024). After two epochs through the training data, our model achieves an **accuracy of 84%** on the held-out test data. [4]

Note that our model has access to a single observation as input, and therefore can not analyze temporal information about the demonstration trajectory. Furthermore, the training and test datasets come from separate demonstration trajectories, so it is also not possible for the model to reason based on trajectory-specific attributes. Hence, for the model to correctly map observations to input, it must find task-specific attributes that generalize to unseen trajectories in the test set. For this to be possible, trajectories from the same task must occupy a common subspace. We conclude that there must be little overlap in the observation space covered by demonstration trajectories corresponding to different tasks.

If one implements a subpolicy network that has access to previous observations in addition to the current observation, this would possibly increase the disjointedness between observation encodings across different tasks. This explains why we observe no significant difference in the cloning loss achieved by a single model mapping observation to action compared to a naive implementation of

---

[4]  While the observations used in LIBERO [16] include an embedding of the current task, we remove this embedding for the purposes of this experiment

temporal variational inference. As explained in Section 4.2, we mitigate this issue by passing a compressed version of the observation space to the subpolicy network, forcing meaningful discrete skills and continuous parameters to be learned in order to minimize behavior cloning loss.

# C  Model Architectures and Hyper-parameters

For our experiments using LIBERO, we use the Resnet-18 image encoder provided in LIBERO [16] for each of the baselines. For MetaWorld, we use the same CNN image encoder used by PRISE. For a fair comparison across methods, we ensure that each algorithm uses the same image encoders, sees the same amount of training data per batch, and is trained/finetuned for the same number of gradient steps. We highlight the architectural details of each baseline below:

## C.1  DEPS

The variational network contains a two-layer bidirectional GRU with a hidden size of 1024. The individual heads for the discrete and continuous parameters both have two layers each, with a hidden size of 1024. For every timestep, the output head for continuous parameters returns an array of dimensions $(|K|, d)$, where $|K|$ is the number of discrete skills (set to 10) and $d$ is the dimensionality of the continuous parameters (set to 4). We then average the outputs of the continuous parameters output heads to get one set of continuous parameters per discrete skill, as highlighted in Section 4.3.

The discrete and continuous networks both consist of one-directional GRU with a hidden size of 1024, followed by two layers of hidden size 1024.

To calculate the one-dimensional compressed state $s'_t$ (see Section 4.2 for notation), we use the provided robot proprioceptive states (8-dimensional in MetaWorld, 9-dimensional in LIBERO) as the feature vector $s_t^{\mathrm{proj}}$. The compression MLP, $f_{\mathrm{compress}}(k_t, z_t)$, consists of two layers with a hidden size of 128. The compressed state is then concatenated with the discrete skill and continuous parameter before being passed through a 2-layer MLP with hidden size 1024 and a Gaussian mixture model head with two intermediate layers and a hidden size of 1024. We use the same Gaussian mixture model implementation as LIBERO [16]. For MetaWorld, we replace the GMM policy head with the deterministic MLP, as has been done by previous work [33].

To calculate the loss during pretraining, in LIBERO we weigh the KL divergence terms corresponding to the discrete skills and continuous variables by 0.5 and 0.01, respectively. The skill parameter norm penalty is weighted by 0.1. For MetaWorld, we scale these hyperparameter down to 0.1, 0.03, and 0.03 respectively, to account for the lower loss magnitudes (due to using a deterministic policy head instead of a GMM).

Batch sizes are chosen to fit a single GPU during pretraining and finetuning. This results in a batch size of 3 during pretraining and a batch size of 2 during finetuning for LIBERO, and batch sizes of 8 during pretraining and 3 during finetuning (i.e. all of the data as we do 3-shot finetuning) for MetaWorld.

We use a learning rate of $3e-4$ and the random seeds 95, 96, 97, 98, and 99.

## C.2  Behavior Cloning

We use the same general setup as described for DEPS. The main changes are that the variational, discrete, and continuous networks are no longer used, and the subpolicy network has access to the complete observation, including images and proprioceptive state. The observation is passed through a 1-layer LSTM with hidden size 1024 before being passed to the same GMM head used above.

## C.3  PRISE

We use the original implementation and hyperparameters provided in Zheng et al. [33], which comes with implementations tuned for LIBERO and MetaWorld. For fair comparison to other algorithms, we make two changes to the provided implementation: (i) we reduce the batch size to fit our computational resources, using batch sizes to have the same amount of data per batch as the other algorithms (ii) we reduce the total number of gradient steps taken to match the other algorithms.

## D Descriptions of Tasks in the Test Set

**LIBERO**  We perform pretraining on the first 80 tasks in LIBERO-90 [16]. The LIBERO-OOD dataset, which tests generalization to unseen tasks consisting of novel objects and environments, corresponds to the last 10 tasks in LIBERO-90. Task IDs and descriptions for this dataset are provided in Table 3. We also test performance on the LIBERO-10, details for which can be found in LIBERO.

**MetaWorld-v2**  We perform pretraining on the first 10 tasks in MetaWorld. Finetuning is then performed on the MW-Vanilla and MW-PRISE datasets, as explained in Section 5.2. Information on the specific tasks used in each dataset is provided in Table 4.

Table 3: Tasks in LIBERO-OOD

| Task Index | Task Description |
|---|---|
| 80 | STUDY SCENE2 pick up the book and place it in the right compartment of the caddy |
| 81 | STUDY SCENE3 pick up the book and place it in the front compartment of the caddy |
| 82 | STUDY SCENE3 pick up the book and place it in the left compartment of the caddy |
| 83 | STUDY SCENE3 pick up the book and place it in the right compartment of the caddy |
| 84 | STUDY SCENE3 pick up the red mug and place it to the right of the caddy |
| 85 | STUDY SCENE3 pick up the white mug and place it to the right of the caddy |
| 86 | STUDY SCENE4 pick up the book in the middle and place it on the cabinet shelf |
| 87 | STUDY SCENE4 pick up the book on the left and place it on top of the shelf |
| 88 | STUDY SCENE4 pick up the book on the right and place it on the cabinet shelf |
| 89 | STUDY SCENE4 pick up the book on the right and place it under the cabinet shelf |

Table 4: Pretraining (left) and finetuning (right) task splits for MetaWorld

| Task Index | Pretraining Task Description |
|---|---|
| 0 | assembly-v2 |
| 1 | basketball-v2 |
| 2 | button-press-topdown-v2 |
| 3 | button-press-topdown-wall-v2 |
| 4 | button-press-v2 |
| 5 | button-press-wall-v2 |
| 6 | coffee-button-v2 |
| 7 | coffee-pull-v2 |
| 8 | coffee-push-v2 |
| 9 | dial-turn-v2 |

| Finetune Dataset | Task Index | Finetune Task Description |
|---|---|---|
| MW-Vanilla | 45 | bin-picking-v2 |
| | 46 | box-close-v2 |
| | 47 | hand-insert-v2 |
| | 48 | door-lock-v2 |
| | 49 | door-unlock-v2 |
| MW-PRISE | 10 | disassemble-v2 |
| | 24 | pick-place-wall-v2 |
| | 37 | stick-pull-v2 |
| | 46 | box-close-v2 |
| | 47 | hand-insert-v2 |

## E Performance on MetaWorld-v2 with Varying Pretraining Amounts

In Section 5 we perform 20 epochs of pretraining on LIBERO and 40 epochs of pretraining on MetaWorld, showing that DEPS outperforms existing baselines in its rapid generalization to diverse datasets of unseen tasks. We also show that DEPS retains its performance advantage on reducing the number of pretraining epochs (using LIBERO-OOD), and, in the case of low pretraining amounts, the performance gap between DEPS and other baselines *increases*. In Table 5, we present results on MetaWorld analogous to those presented in Table 1, but with half as much pretraining (i.e. 20 epochs). As can be seen DEPS retains in its performance advantage, suggesting that it is robust to different pretraining compute budgets.

## F DEPS with Different Amounts of State Compression

One may ask whether compressing the observation to 1D is overly aggressive.

As outlined in Section 4.2, we believe this is not the case, as a parameterized skill can be viewed as a family of related parameterized trajectories. Each trajectory is defined by a choice of continuous parameterization, and given a fixed trajectory, a 1D state is sufficient as one only needs an "index" into the trajectory (Figure 3). As the dimensionality of the state embedding increases, we expect

Table 5: Evaluation results on MetaWorld with half as much pretraining (20 epochs). All results are averaged across 5 seeds

| Evaluation Set | Algorithm | Mean Success | Mean Highest Success |
|---|---|---|---|
| MW-Vanilla | DEPS | **0.40** $\pm$ 0.02 | **0.58** $\pm$ 0.04 |
| | PRISE | 0.19 $\pm$ 0.06 | 0.31 $\pm$ 0.08 |
| | BC | 0.35 $\pm$ 0.02 | 0.50 $\pm$ 0.03 |
| MW-PRISE | DEPS | **0.30** $\pm$ 0.04 | **0.51** $\pm$ 0.05 |
| | PRISE | 0.07 $\pm$ 0.03 | 0.18 $\pm$ 0.06 |
| | BC | 0.24 $\pm$ 0.01 | 0.42 $\pm$ 0.03 |

more information to be encoded in the state embedding, which implies (i) lower overlap in the embedded state-space across tasks and (ii) less reliance on the skill parameters to encode meaningful information, resulting in poorer performance in unseen tasks/environments.

To validate this intuition, we ablate the performance of DEPS against versions of DEPS using 2D and 3D state compression instead. Additionally, we compare against DEPS (Uncompressed), which uses a subpolicy implementation that takes the history of robot proprioceptive states, which are then passed through an LSTM (without any compression) before being concatenated with the current discrete skill and continuous parameter and passed through an output head.

As can be seen in Table 6, DEPS significantly outperforms each of the baselines.

Table 6: Performance of parameterized skills with different compression dimensions. Results are on LIBERO-OOD. 1D state compression results taken from Table 1. Results for 2D and 3D compression and DEPS (Uncompressed) are averaged over 3 seeds.

| Compression Dimension | Mean Success Rate | Mean Highest Success Rate |
|---|---|---|
| DEPS | **0.34** $\pm$ 0.08 | **0.66** $\pm$ 0.12 |
| DEPS (2D state) | 0.13 $\pm$ 0.12 | 0.30 $\pm$ 0.18 |
| DEPS (3D state) | 0.05 $\pm$ 0.04 | 0.17 $\pm$ 0.11 |
| DEPS (Uncompressed) | 0.19 $\pm$ 0.05 | 0.48 $\pm$ 0.03 |

## G  DEPS with Different Limits on the Maximum Number of Discrete Skills, $K$

In all our experiments in Section 5, we set the maximum number of discrete skills $K$ to 10. This is simply meant to be an upper bound on the number of discrete skills learnable by DEPS. To demonstrate the robustness of DEPS to different values of this hyperparameter, we ablate its performance with a different value of this same hyperparameter ($K = 20$) in Table 7. As can be seen, setting $K = 20$ in fact *significantly improves* the performance of DEPS. It is possible that the performance of DEPS might increase even more significantly with a thorough hyperparameter sweep.

Table 7: Performance of DEPS with different values of $K$. Results for $K = 10$ taken from Table 1. Results for $K = 20$ are averaged across 3 seeds.

| K | Mean Success Rate | Mean Highest Success Rate |
|---|---|---|
| 10 (from paper) | 0.34 $\pm$ 0.08 | 0.66 $\pm$ 0.12 |
| 20 | **0.47** $\pm$ 0.01 | **0.74** $\pm$ 0.05 |

## H  On the Importance of *Parameterized* Skills

As discussed in Section 1, we believe that learning purely discrete or continuous skills possesses inherent limitations. DEPS' two tiered design combines the benefits of these two approaches, providing the representation power of continuous parameters with the structured and interpretable

nature of discrete skills. In Table 8, we we compare the empirical performance of DEPS against approaches using only discrete or continuous parameters, showing that DEPS outperforms both settings.

Table 8: Performance comparison between DEPS and ablated versions with only discrete skills or only continuous parameters. Results are on LIBERO-OOD. Results for DEPS are taken from Table 1. Results for discrete-only and continuous-only are averaged across 3 seeds.

| Method | Mean Success | Mean Max Success |
|---|---|---|
| DEPS (results from paper) | $\mathbf{0.34} \pm 0.08$ | $\mathbf{0.66} \pm 0.12$ |
| Discrete-only | $0.06 \pm 0.02$ | $0.16 \pm 0.01$ |
| Continuous-only | $0.15 \pm 0.16$ | $0.32 \pm 0.30$ |

## I   On the Impact of Evaluation Horizon

The choice to evaluate DEPS after a limited number of finetuning gradient steps (500) is intentional and reflects the core motivation of our work: to learn transferable representations that enable rapid adaptation to new tasks. We believe this evaluation protocol serves as a stronger test of generalization ability as it demonstrates that DEPS learns genuinely transferable skills rather than simply learning the downstream tasks from scratch.

For completeness, we provide additional results comparing the performance of DEPS and the BC baselines after 5x as much finetuning (i.e. 2500 gradient steps). We perform rollouts after every 250 gradient steps of finetuning and report the Mean Success Rate and the Mean Highest Success Rates. As can be seen, DEPS maintains its performance advantage, with the gap between DEPS and BC actually *increasing*

Table 9: Performance comparison between DEPS and BC on LIBERO-OOD with 5x as many finetuning gradient steps. Results are averaged across 3 seeds.

| Method | Mean Success | Mean Max Success |
|---|---|---|
| DEPS | $\mathbf{0.49} \pm 0.05$ | $\mathbf{0.75} \pm 0.02$ |
| BC | $0.20 \pm 0.04$ | $0.35 \pm 0.04$ |

## J   Limitations

DEPS shows significant performance improvements in rapid generalization abilities in diverse and challenging environments. However, there are some important limitations to consider:

**Generalization and accuracy tradeoffs**   As can be seen in Table 1, the performance of DEPS is not as strong on LIBERO-10, which consists largely of *in-distribution* but longer-horizon tasks, compared to other held-out task sets. Given the in-distribution nature of LIBERO-10, it makes sense that parameterized skills will offer less of a performance improvement as there is no need to generalize to new tasks or environments. However, it is also possible that our choice to aggressively compress observations to one dimension before passing them to the subpolicy network, which enables generalization, potentially comes at the cost of perfect reconstruction of in-distribution long-horizon demonstration trajectories. The observed tradeoff may be acceptable in many practical robotics scenarios, where learned skills from demonstrations typically serve as a starting point for further refinement using RL. In such cases, the initial generalization capability is more valuable than perfect execution on specific, lengthy sequences.

We envision several promising approaches to address this limitation while preserving the generalization benefits of our method. For instance, one may train a residual policy that has access to the full-observation space in addition to the current discrete skill and continuous parameter, correcting the output of the original subpolicy network. We leave a more rrigorous evaluation of this potential shortcoming and its remedies to future work.

**Reliance on proprioceptive robot states**    To implement the compression strategy outlined in Section 4.2, we use the robot's proprioceptive state, which is provided in both LIBERO and MetaWorld, as the feature vector $s_t^{\text{proj}}$. This provides a compact, low-dimensional representation of the robot's state, while excluding information about the environment and goal, which must then be encoded in the discrete skills and continuous parameters. While we expect our method to generalize to environments where proprioceptive state is not provided, we leave an evaluation of this to future work.

**Method complexity**    Compared to naive behavioral cloning, DEPS is more complex to implement and has more hyperparameters, resulting in more hyperparameter tuning. While we show that DEPS has clear performance advantages, it may not be appropriate for simpler environments or narrow task distributions requiring less generalization.

## K    Societal Impacts and Potential for Misuse

We do not perform studies involving human subjects, and properly attribute datasets and software used to their authors. The research presented, which focused on enabling robots to learn parameterized skills from demonstrations, has the potential for significant positive societal impact. These include enhanced automation in industrial and domestic settings, improved assistive technologies for individuals needing support, and safer execution of tasks in hazardous environments. On the other hand, our method inherits the common potential for misuse of robotics research, such as autonomous weapons, labor displacement, and economic disruption, but these potentials are not unique to this paper.

## L    Further Visualization of Learned Skills

We provide more visualizations of the discovered parameterized skills on our website: sites.google.com/view/parameterized-skills.

