# OpenReview forum: "Learning Parameterized Skills from Demonstrations"
_NeurIPS.cc/2025/Conference — NeurIPS 2025 poster_

### Official Review · Reviewer_KY1m · 2025-06-21

**Clarity:** 3
**Significance:** 3
**Originality:** 3
**Rating:** 5
**Confidence:** 3

**Summary:**

The authors propose a novel skill discovery and skill-based imitation learning algorithm that learn parameterized skills from demonstrations.  The parameterized skills are a policy conditioned on both a discrete input that acts as a skill selector and a continuous input that acts as the parameter to that skill.  The discrete skill selector, continuous parameter selector, and low level policy are learned end to end, VAE-style using BC loss, with a variational distribution on the discrete and continuous skill variable given a task trajectory.  In order to handle degenerate solutions and ensure the skill variables actually encode useful information, the authors propose to compress the state input to the low-level policy into a single dimension.   They evaluate their model's ability to generalize to novel tasks with a fixed finetuning budget after being pretrained on demonstrations from the same domain in LIBERO and MetaWorld task suites and compare favorably to BC and an LLM-style skill discovery method.

**Questions:**

1. How is your. method finetuned?
2. How do you ensure both $k$ and $z$ encode useful and distinct information?

**Ethical Concerns:**

["NO or VERY MINOR ethics concerns only"]

**Final Justification:**

I am recommending a score of 5.  This is a well-written paper with clear contributions and novelty.  I appreciate the authors' ablations on the state compression from the rebuttal which provided additional insight into their method.

**Limitations:**

Yes, although I would prefer if they were summarized in the main paper.

**Paper Formatting Concerns:**

No formatting concerns.

**Quality:**

3

**Strengths And Weaknesses:**

Strengths:
1. Novel and well-motivated method idea, particularly in using state compression in the low level policy.
2. Strong experimental results
3. Clear and well-written paper

Weaknesses:
1. Why are there no comparisons to other variational skill discovery methods like [1], [2]?
2. This work is missing some ablations, especially ablating having both discrete and continuous skill latent variables instead of just one or the other.  I am curious how important it is to actually have both or could the skill policy be successful parameterized with just a continuous variable.  It would also be helpful to ablate the state compression mechanism to see how much compression is necessary to learn meaningful skills.

[1] Joshua Achiam, Harrison Edwards, Dario Amodei, Pieter Abbeel. Variational Option Discovery Algorithms
[2] Karl Pertsch, Youngwoon Lee, Joseph J. Lim. Accelerating Reinforcement Learning with Learned Skill Priors.

---

> ### Author Rebuttal · Authors · 2025-07-31
>
> Thank you for the positive and thoughtful comments. We address your questions below.
>
> **Q1:** Why are there no comparisons to other variational skill discovery methods like [1], [2]?
>
> **A1:** We center our comparisons against PRISE, as it is a recent and performant work; we thus utilize the same environments (LIBERO and Metaworld) for fair, standardized evaluations.  To the best of our knowledge, there are no available, performant implementations or existing evaluations for [1] and [2] that have been tuned for LIBERO and Metaworld.
>
> **Q2:** This work is missing some ablations, especially ablating having both discrete and continuous skill latent variables instead of just one or the other. I am curious how important it is to actually have both or could the skill policy be successful parameterized with just a continuous variable. It would also be helpful to ablate the state compression mechanism to see how much compression is necessary to learn meaningful skills.
>
> **A2:** Thank you for raising this important suggestion. We have run the ablations you have recommended. We present the results here:
>
> 1. **Only discrete or continuous variables:** the following table compares the performance on LIBERO-OOD between DEPS (results from the paper), (ii) only discrete skills (iii) only continuous parameters. The results for (ii) and (iii) are averaged across 3 seeds
>
>    | Method | Mean Success | Mean Max Success |
>    |--------|--------------|------------------|
>    | DEPS (results from paper) | 0.34 ± 0.08 | 0.66 ± 0.12 |
>    | Discrete only | 0.06 ± 0.02 | 0.16 ± 0.01 |
>    | Continuous only | 0.15 ± 0.16 | 0.32 ± 0.30 |
>
>    DEPS outperforms both settings. We find these results to be consistent with our presented motivation for a two-tiered design in Section 1. Learning only discrete skills can prevent generalization over a smoothly varying distribution of tasks. Similarly, learning purely continuous parameters leads to less structured and interpretable representations.
>
> 2. **Amount of state compression:** To validate the importance of 1D state compression to DEPS’ strong performance, we compare the mean success rates and the mean highest success rates on the LIBERO-OOD test setting for DEPS with 1D state compression (results from the paper) against DEPS with 3D state compression. The results for 3D state compression are averaged across 3 seeds.
>
>
>    | Compression Dimension | Mean Success Rate | Mean Highest Success Rate |
>    |----------------------|-------------------|---------------------------|
>    | 1D (from paper)      | 0.34 ± 0.08      | 0.66 ± 0.12              |
>    | 3D                   | 0.05 ± 0.04      | 0.17 ± 0.11              |
>
>    The dramatic performance drop from 1D to 3D compression (0.34 → 0.05 mean success rate) suggests that aggressive compression to a single dimension is crucial to DEPS’ performance.
>
> **Q3:** How is your method finetuned?
>
> **A3:** We outline the finetuning procedure in Appendix A5. While we use the same general approach as in the pretraining phase, we no longer utilize the variational network. Instead, we now sample the discrete skill, $k$, and the continuous parameter, $z$, from the autoregressive policies $\pi^K$ and $\pi^Z$. This is done to make the finetuning process align with the process used to sample actions during inference.
>
> **Q4:** How do you ensure both $k$ and $z$ encode useful and distinct information?
>
> **A4:** Thank you for the insightful question! Empirically, the first ablation in A2 validates the need for both $k$ and $z$.
> One of DEPS’ main goals is to ensure that $k$ and $z$ encode useful information. As outlined in Appendix A1, during pretraining, the variational network predicts continuous parameters per discrete skill for each forward pass over a given trajectory (instead of predicting a new continuous parameterization at each time step, which could lead to overfitting).  Additionally, to prevent the continuous parameters from overfitting to specific tasks, we introduce the $\mathcal{L}_{\text{norm}}$ regularization term (Appendix A1), which bounds the range of the learned continuous parameters. This limits the amount of information encoded in the discrete skill and continuous parameters, forcing the 1D state to encode meaningful information as an “index” into parameterized skill trajectories.  This is validated in Figure 8, where the learned latent 1D states vary monotonically within discrete skills, with discontinuities when the discrete skill switches. The strong performance of DEPS on out-of-distribution downstream tasks (Table 1, LIBERO-OOD) further supports our hypothesis that DEPS learns meaningful abstractions instead of overfitting to $k$ or $z$ values. This is particularly evident in low-data regimes (Table 1, LIBERO-3-shot) and low pretraining settings (Table 2).
>
> We provide several lines of evidence that $k_t$ and $z_t$ do encode skill-relevant and distinct information:
>
> 1. **Consistent application of discrete skills (Section 5.5 and Figure 7):** We find that DEPS discovers semantically meaningful discrete skills corresponding to primitive behaviors such as grasping objects, opening doors, etc. Importantly, we find that the discovered skills are applied *consistently* across tasks. An example can be seen in Figure 7, where the same ``grasp_object’’ skill is used to pick butter in a cabinet environment (top) and pick a mug in a kitchen environment (bottom). This is not a cherry-picked example; we can release an annotated version of the entire LIBERO dataset across multiple tasks, showing that learned discrete skills consistently hold across distinct tasks and environments.
>
> 2. **Smooth variation in policy on varying continuous parameters (Section 5.5):** We find that for a given discrete skill, slight modifications in the continuous parameter result in smooth variations in the resulting policy. For example, modifying the continuous parameters for a discrete skill used to turn a knob smoothly varies the location at which the knob turning action is executed. Visualizations of this are provided on our website (which was already included at the time of submission; the URL is in section 5.5).
>
> Thank you again for your constructive feedback, which has helped improve the clarity and position of our paper. Please let us know if you have any additional questions.

---

> > ### Comment · Reviewer_KY1m · 2025-08-01
> >
> > Thank you for the thorough response and the added ablations.  I found the state compression ablation especially interesting since it shows that compressing the state to a single dimension is a crucial component of this method and may also be a key mechanism that ensures a good $k$ and $z$ space.  I will be maintaining my score.

---

### Official Review · Reviewer_9spN · 2025-06-24

**Clarity:** 3
**Significance:** 2
**Originality:** 2
**Rating:** 3
**Confidence:** 4

**Summary:**

This paper introduces **DEPS**, an end-to-end framework that jointly discovers discrete skills and continuous parameters from demonstrations, trained via temporal variational inference augmented with information-bottleneck tricks. Empirically, DEPS outperforms BC and PRISE on LIBERO and MetaWorld benchmarks under few-shot fine-tuning.

**Questions:**

1. **How is *K* chosen?** Please describe any heuristics, cross-validation or sensitivity analysis; current text implies it is fixed but gives no guidance.
2. **Single-latent ablation** – add experiments with (i) only discrete skills and (ii) only continuous parameters to prove the necessity of the two-tier latent design.
3. **End-to-end baselines** – include a diffusion-policy or transformer-BC baseline that maximises πᵃ directly without latent regularisation to test whether πᵃ+{πᴷ,πᶻ} is truly superior.
4. **Risk of latent shortcuts** – with 1-D state input the sub-policy may over-fit to *k* or *z* values rather than learning robust motor primitives. Can you visualise latent distributions to rule this out or try less aggressive compression?
5. **Evaluation horizon** – report performance after full convergence on the new task. Including both “rapid-adapt” and “fully-finetuned” results would clarify absolute capability and asymptotic ceiling.

**Ethical Concerns:**

["NO or VERY MINOR ethics concerns only"]

**Final Justification:**

The paper's experimental setting is uncommon and does not include more mainstream experimental settings, and lacks some more common baseline comparisons.

**Limitations:**

yes

**Paper Formatting Concerns:**

No formatting issues.

**Quality:**

3

**Strengths And Weaknesses:**

**Strengths**

* **Principled objective** – derives a tractable ELBO with exact KL terms for both categorical and Gaussian latents.
* **Degeneracy mitigation** – the 1-D state compression and latent-conditioned sub-policy convincingly force reliance on the latents, and ablations show sizable gains.
* **Interpretability** – visual segmentations reveal intuitive “grasp / move / release / close” skills reused across tasks; continuous parameters smoothly modulate grasp poses.

**Weaknesses**

* **Hyper-parameter sensitivity** – the number of discrete skills *K* is fixed a-priori; no heuristic or validation study is given.
* **Two-level latent justification weak** – authors assert both *k* and *z* are needed but provide no ablation vs. single-latent variants.
* **Baselines limited** – modern end-to-end policies (e.g., Diffusion Policy) are absent; PRISE is the only hierarchical competitor.
* **Evaluation setup peculiar** – success is measured after ≤ 500 gradient steps instead of on a fully converged policy, making absolute numbers hard to interpret.
* **Compression extremity** – projecting proprioception to a single scalar risks under-representation; possible shortcut utilisation by latents is not checked.

---

> ### Author Rebuttal · Authors · 2025-07-31
>
> Thank you for your thoughtful and thorough review. We address your questions below.
>
> **Q1: How is K chosen?** Please describe any heuristics, cross-validation, or sensitivity analysis; the current text implies it is fixed but gives no guidance.
>
> **A1:** You raise an important question about the hyperparameter sensitivity of DEPS. In our experiments, we set the maximum number of skills (K) to the same value (10) across all datasets and environments. The value of K is meant to simply be a large number that sets an upper bound on the number of discrete skills that can be learned by DEPS. This is a common technique used in previous skill learning papers (e.g. [1] and [2]). In practice, we find that the number of discrete skills learned by DEPS varies across training datasets, but is below K on average. Because of this, we believe DEPS is robust to higher values of this hyperparameter, and therefore did not prioritize sweeping different values of K.
>
> To validate the robustness of DEPS to different values of K, we present some new results of DEPS on LIBERO-OOD setting K to be twice as large (20). Results for K=20 are averaged across 3 seeds.
>
> | K | Mean Success Rate | Mean Highest Success Rate |
> |----------------------|-------------------|---------------------------|
> | 10 (from paper)      | 0.34 ± 0.08      | 0.66 ± 0.12              |
> | 20                   | 0.47 ± 0.01      | 0.74 ± 0.05              |
>
> As can be seen, setting K=20 in fact improves the performance of DEPS. In general, we find that DEPS is highly resilient to different hyperparameter settings. Due to limited compute availability, we did not do much hyperparameter tuning on DEPS, and mostly used the same hyperparameter settings across datasets/environments.
>
> [1] Learning Options via Compression. Jiang et. al. NeurIPS 2022.
> [2] Language Guided Skill Learning with Temporal Variational Inference. Fu et al. ICML 2024
>
> **Q2: Single-latent ablation** – add experiments with (i) only discrete skills and (ii) only continuous parameters to prove the necessity of the two-tier latent design.
>
> **A2:** Thank you for the suggestion. We have added the new experiments, and the following table compares the performance on LIBERO-OOD between DEPS (results from the paper), (ii) only discrete skills (iii) only continuous parameters. The results for (ii) and (iii) are averaged across 3 seeds
>
> | Method | Mean Success | Mean Max Success |
> |--------|--------------|------------------|
> | DEPS (results from paper) | 0.34 ± 0.08 | 0.66 ± 0.12 |
> | Discrete only | 0.06 ± 0.02 | 0.16 ± 0.01 |
> | Continuous only | 0.15 ± 0.16 | 0.32 ± 0.30 |
>
> DEPS outperforms both settings. We find these results to be consistent with our presented motivation for a two-tiered design in Section 1. Learning only discrete skills can prevent generalization, while learning purely continuous parameters can lead to less structured and interpretable representations. On the other hand, DEPS learns semantically interpretable discrete skills whose policies vary smoothly with the continuous parameter value. Visualizations can be found at our website (which was already included at the time of submission; the URL is in section 5.5).
>
> **Q3: End-to-end baselines** – include a diffusion-policy or transformer-BC baseline that maximises πᵃ directly without latent regularisation to test whether πᵃ+{πᴷ,πᶻ} is truly superior.
>
> **A3:** We would like to clarify that the BC baseline included in all our experiments (e.g., in Tables 1 and 2)  is end-to-end. To ensure a fair comparison between DEPS and the BC baseline, we use the same network architecture (LSTM) for both. Therefore, we don't compare diffusion-policy or transformer-BC as the difference in the fundamental network structure makes direct comparison difficult. We focus on DEPS as a general *framework* for learning parameterized skills from demonstrations; it can be easily extended to diffusion/transformer based policies, and we leave such architecture sweeps as future work.
>
> **Q4: Risk of latent shortcuts** – with 1-D state input the sub-policy may over-fit to $k$ or $z$ values rather than learning robust motor primitives. Can you visualise latent distributions to rule this out or try less aggressive compression?
>
> **A4:** **(On latent shortcuts)** You raise an important point about the potential to overfit to $k$ or $z$ values. One of DEPS’ main goals is to prevent this from occurring. As outlined in Appendix A1, during pretraining, the variational network predicts a fixed continuous parameter per discrete skill for each forward pass over a given trajectory (instead of predicting a new continuous parameterization at each time step, which could lead to overfitting). Furthermore, to prevent the continuous parameters from overfitting to specific tasks, we introduce the $\mathcal{L}_{\text{norm}}$ (Appendix A1) regularization term, which bounds the range of the learned continuous parameters. This limits the amount of information encoded in the discrete skill and continuous parameters, forcing the 1D state to encode meaningful information as an “index” into parameterized skill trajectories.  This is validated in Figure 8, where the learned latent 1D states vary monotonically within discrete skills, with discontinuities when the discrete skill switches. The strong performance of DEPS on out-of-distribution downstream tasks (Table 1, LIBERO-OOD) further supports our hypothesis that DEPS learns meaningful abstractions instead of overfitting to $k$ or $z$ values. This is particularly evident in low-data regimes (Table 1, LIBERO-3-shot) and low pretraining settings (Table 2).
>
> **(On visualizing latent distributions)** In Section 5.5, we provide visualizations showing that discovered discrete skills correspond to primitive behaviors such as grasping objects, opening doors, etc. Importantly, we find that the discovered skills are applied consistently across tasks. An example can be seen in Figure 7, where the same ``grasp_object’’ skill is used to pick butter in a cabinet environment  (top) and pick a mug (bottom) in a kitchen environment (bottom). This is not a cherry-picked example; we can release an annotated version of the entire LIBERO dataset across multiple tasks, showing that learned discrete skills consistently hold across distinct tasks and environments. Additionally, we have visualizations that show that for a given discrete skill, there is significant overlap in the continuous parameters chosen for different tasks, ruling out the possibility of overfitting to $z$. This further validates that DEPS learns robust motor primitives. We are happy to update the project website with such visualizations for the reviewers' interest during the discussion period, provided the AC gives approval or clarification on the website update policy; regardless, we will include such visualizations in the final camera-ready.
>
>
> **(On compression aggression)** To validate the importance of 1D state compression to DEPS’ strong performance, we compare the mean success rates and the mean highest success rates on the LIBERO-OOD test setting for DEPS with 1D state compression (results from the paper) against DEPS with 3D state compression. The results for 3D state compression are averaged across 3 seeds.
>
>
> | Compression Dimension | Mean Success Rate | Mean Highest Success Rate |
> |----------------------|-------------------|---------------------------|
> | 1D (from paper)      | 0.34 ± 0.08      | 0.66 ± 0.12              |
> | 3D                   | 0.05 ± 0.04      | 0.17 ± 0.11              |
>
> The dramatic performance drop from 1D to 3D compression (0.34 → 0.05 mean success rate) suggests that aggressive compression to a single dimension is crucial to DEPS’ performance.
>
>
> **Q5: Evaluation horizon** – report performance after full convergence on the new task. Including both “rapid-adapt” and “fully-finetuned” results would clarify absolute capability and asymptotic ceiling.
>
> **A5:** Our choice to evaluate after limited gradient steps (500 steps) is intentional and reflects the core motivation of our work: to learn transferable representations that enable rapid adaptation to new tasks. We believe this evaluation protocol serves as a stronger test of generalization ability as it demonstrates that DEPS learns genuinely transferable skills rather than simply learning the downstream tasks from scratch.
>
> To evaluate DEPS’ asymptotic ceiling, we’ve performed additional experiments comparing the performance of DEPS and BC on the LIBERO-OOD evaluation setting with 5x as much finetuning (i.e. 2500 gradient steps). We perform rollouts after every 250 gradient steps of finetuning and report the Mean Success Rate and the Mean Highest Success Rates. The following results are averaged across 3 seeds:
>
> | Method | Mean Success Rate | Mean Highest Success Rate |
> |--------|-------------------|---------------------------|
> | DEPS | 0.49 ± 0.05 | 0.75 ± 0.02 |
> | BC | 0.20 ± 0.04 | 0.35 ± 0.04 |
>
> We find that our method is still better than the baseline, and the gap between DEPS and BC actually increases.
>
> We appreciate the constructive feedback, which has helped improve the experimental results in our paper. Please let us know if you have any further questions, and we hope you will consider raising your score in light of our response and additional results.

---

> > ### Comment · Reviewer_9spN · 2025-08-05
> >
> > Thanks for the detailed responses. The authors have addressed some of my concerns. However, my reservations remain due to the lack of some strong baselines and the fact that the experiments do not cover all four standard settings of the LIBERO benchmark. Therefore, I will keep my current score.

---

> > > ### Author Response · Authors · 2025-08-07
> > >
> > > Thank you for your continued engagement. We are glad to have been able to address some of your previously listed concerns, and seek to offer additional clarifications below:
> > >
> > > **On including additional baselines**, we would like to emphasize that one of our baselines, PRISE, is a recent, performant algorithm (ICML 2024, Oral). Additionally, we believe that including a transformer-based BC baseline may not align directly with the motivation of our paper. We focus on DEPS as a general framework to learn parameterized skills, and therefore use the same underlying neural network architecture (LSTM) for both DEPS and the BC baseline for fair comparison. This allows us to isolate the effect of parameterized skill abstraction from architectural improvements.
> > >
> > > **On LIBERO evaluation datasets,** we would like to clarify that of the four datasets provided by default with LIBERO (LIBERO-object, LIBERO-spatial, LIBERO-goal, and LIBERO-100), only LIBERO-100 has a pretraining and evaluation task split (where LIBERO-90 is the pretraining dataset and LIBERO-10 is the evaluation set), and we do include results on this test setting in our results (LIBERO-10 in Table 1). The other three datasets each consist of only 10 tasks with **no pretraining and finetuning task splits**, as these datasets have been curated for continual learning. More importantly, they have very narrow distribution shifts and diversity (e.g. LIBERO-object consists of the same environment and task, but with varying object positions). For the above reasons, we only evaluate on LIBERO-10 and LIBERO-OOD.  Prior works that rely on a pretraining/finetuning procedure (not necessarily limited to skill learning) **also** frequently omit evaluations on these three smaller LIBERO datasets (e.g, PRISE).
> > >
> > > Nevertheless, we believe that the *heart of the reviewer’s concern* lies not in details regarding LIBERO, but on the *thoroughness of our evaluations*.  We thus would like to emphasize that for evaluating skill learning, our evaluations go considerably above and beyond to be as comprehensive as possible. To this end, in addition to our evaluations on LIBERO-10 and LIBERO-OOD, we include 3-shot finetuning results on LIBERO to show that DEPS is performant under different quantities of finetuning data. We also evaluate DEPS under different pretraining amounts (Table 2) and different finetuning amounts (Figure 6) to show that DEPS is performant under varying pretraining/finetuning budgets. For each of the evaluation settings outlined above, we include two measures of success, measuring both average performance (Mean Success), **and** the highest performance across finetuning epochs (Mean Highest Success), whereas only the latter metric is used by prior works such as PRISE and LIBERO.
> > >
> > > In summation, we have strived to evaluate DEPS under a diverse and rigorous set of evaluations, including (i) different environments and task splits, (ii) different pretraining budgets, (iv) different finetuning budgets, (iv) different amounts of finetuning data (v) different measures of success.
> > >
> > > We thank you for the continued discussions and helpful suggestions. We are happy to address any further questions you may have; please do not hesitate to let us know.

---

> > > > ### Comment · Reviewer_9spN · 2025-08-08
> > > >
> > > > Thank you for your response. I have a follow-up question to ensure I fully understand the experimental procedure.
> > > >
> > > > Could you please provide a more detailed, step-by-step description of the entire training and testing pipeline for the experiments on the LIBERO benchmark(LIBERO-OOD, LIBERO-10 and LIBERO-3-shot)?
> > > >
> > > > For instance, this would ideally include details on:
> > > > - The data used for any initial training phase(s).
> > > > - The specific training procedure(s) and key hyperparameters.
> > > > - A description of any subsequent stages, such as fine-tuning or adaptation, if they were used.
> > > > - The final evaluation protocol on the test set.
> > > >
> > > > A clear, end-to-end overview of this process would be very helpful for me to fully assess the methodology.

---

> > > > > ### Author Response · Authors · 2025-08-09
> > > > >
> > > > > Thank you for the question. We are happy to outline our experimental procedure step-by-step for the reviewer’s interest:
> > > > >
> > > > > **Data:**
> > > > >
> > > > > The original full dataset provided is LIBERO-100 and the standard setting splits it into a pretraining set (LIBERO-90), and a testing set (LIBERO-10). One appealing application for HRL is rapid adaptation to OOD tasks. Since LIBERO-10 mostly consists of in-distribution tasks (as concatenations of pretraining tasks), we further split LIBERO-90 into LIBERO-OOD (which consists of 10 out-of-distribution tasks) to explicitly highlight the benefits of skill learning for task generalization. The remaining 80 tasks in LIBERO-90 are used for pretraining.
> > > > >
> > > > > **Step 1: Pretraining (Section 5.2)**
> > > > >
> > > > > We use the same pretraining dataset consisting of 80 tasks from LIBERO-90 for all LIBERO benchmarks (LIBERO-OOD, LIBERO-10 and LIBERO-3-shot). For each pretraining task, we use all 50 original demonstration trajectories provided by LIBERO. For our main results (Table 1), we perform 20 passes over the pretraining dataset (for comprehensiveness, results with other amounts of pretraining are provided in Table 2).
> > > > >
> > > > > To pretrain DEPS, we use the objective in Equation 4. Implementation details for DEPS can be found in Appendix A.
> > > > >
> > > > > **Step 2: Finetuning**
> > > > >
> > > > > We use the same finetuning procedure for each benchmark. Specifically, for each task in the benchmark, we perform 500 gradient steps of finetuning using the provided demonstrations for the task. As explained in Section 5, after every 50 gradient steps of finetuning on a given task, we perform an evaluation by conducting 20 rollouts, where each rollout is assigned a binary reward based on successful completion of the task. The average success rate from the 20 rollouts is calculated for each of the 10 evaluations performed for the given task (i.e. after 50, 100, …, 500 gradient steps, respectively).
> > > > >
> > > > > Our evaluations on LIBERO-10 consist of the original dataset provided by LIBERO, which contains 50 demonstrations for each of the 10 tasks. However, since tasks in LIBERO-10 are predominantly concatenations of tasks in LIBERO-90, we also use the LIBERO-OOD dataset, which involves interacting with unseen objects/environments (making this a stronger test of generalization). For each task in LIBERO-OOD, we use the 50 provided demonstrations during evaluation. The LIBERO-3-shot dataset consists of the same tasks as LIBERO-OOD, but we only use the first 3 demonstration trajectories provided by LIBERO. Hence, compared to LIBERO-OOD, LIBERO-3-shot evaluates the ability of a given algorithm to generalize under a lower availability of demonstration trajectories on the unseen task (as explained in Section 5.2).
> > > > >
> > > > > For each algorithm, we perform pretraining+finetuning on 5 seeds. Given a benchmark (i.e., LIBERO-10, LIBERO-OOD, or LIBERO-3-shot), we hence conduct 10 evaluations for each of the 10 tasks and for each of the 5 seeds.
> > > > >
> > > > > **Step 3: Evaluation**
> > > > >
> > > > > We report the collected evaluations using the following metrics (as explained in Section 5):
> > > > >
> > > > >  - Mean Highest Success: For each task+seed, we take the highest success rate from the 10 evaluations, and then average this value across all 10 tasks and 5 seeds. This is the metric used by previous work, such as PRISE and LIBERO, and measures performance under the best number finetuning gradient steps out of the 10 evaluations.
> > > > >
> > > > >  - Mean Success: The average success rate across all 10 evaluations (each of which consists of 20 rollouts) for each of the 10 tasks and 50 seeds. This value is hence the average of 10,000 datapoints and is meant to measure the consistent performance of an algorithm.
> > > > >
> > > > > We also report the trend in success rate as a function of the number of finetuning gradient steps in Figure 6. As can be seen, DEPS shows superior performance irrespective of the number of finetuning gradient steps.
> > > > >
> > > > > **Key Hyperparameters Used:**
> > > > >
> > > > > For fair comparison between the evaluated algorithms, we use the same ResNet-18 image encoder implementation (originally provided in LIBERO) and ensure that each method sees the same amount of training data per batch and is trained/finetuned for the same number of gradient steps.
> > > > >
> > > > > For DEPS, we set the maximum number of discrete skills $K$ to be 10 (we ablate this in our response above) and the dimension of the compressed state to 1 (motivation in Section 4.2 and ablation in our general comment). We use a hidden size of 1024 and a learning rate of 3e-4. The BC baseline uses an LSTM with hidden size 1024 and access to the full image embedding generated by the ResNet-18 encoder (i.e. no compression unlike DEPS). For PRISE, we use the original implementation by the authors, which has the same hidden size (1024). Details on the hyperparameters used can be found in Appendix C.
> > > > >
> > > > > We thank you for your continued engagement and hope our response answers your questions.

---

> > > > > > ### Comment · Reviewer_9spN · 2025-08-09
> > > > > >
> > > > > > Thank you for the detailed response. The experimental procedure is much clearer now. However, I have two minor follow-up questions:
> > > > > >
> > > > > > 1. In "Step 1: Pretraining," you mention that "we perform 20 passes over the pretraining dataset." Could you please clarify what "20 passes" means in this context? For instance, does it refer to training for 20 full epochs on the pretraining dataset? I just want to confirm the exact operation during this pretraining stage.
> > > > > >
> > > > > > 2. You state that LIBERO-OOD and LIBERO-3-shot are "a stronger test of generalization" compared to LIBERO-10. However, looking at the results, why are the success rates on these two settings considerably higher than on LIBERO-10? Intuitively, one might expect a lower success rate for more challenging generalization tasks. Could you offer some insight into this interesting phenomenon?
> > > > > >
> > > > > > Thank you again for your engagement.

---

> ### Author Response · Authors · 2025-08-09
>
> 1. Yes, 20 passes refer to 20 full epochs of pretraining on the LIBERO dataset (80 tasks).
> 2. The reviewer indeed raises an interesting point. As we partially explained in Appendix G, the success rate discrepancy can be explained by the inherent design of the datasets.  The key feature of LIBERO-10 is that it consists largely of **in-distribution** but **longer-horizon** tasks, while LIBERO-OOD consists of **novel objects and environments** that the agent has never seen during pretraining and the horizon is similar compared to the pretrained tasks (10 unseen tasks from LIBERO-90, which is shorter-horizon). The performance of DEPS is not as strong on LIBERO-10 (**but still outperforms alternative baselines**), compared to LIBERO-OOD. A potential reason for this is our choice to aggressively compress observations to one dimension before passing them to the subpolicy network, which enables generalization, potentially at the cost of perfect reconstruction of long-horizon demonstration trajectories. The observed tradeoff may be acceptable in many practical robotics scenarios, where learned skills from demonstrations typically serve as a starting point for further refinement using RL. In such cases, the initial generalization capability is more valuable than perfect execution on specific, lengthy sequences. We believe that presenting LIBERO-OOD/LIBERO-3-Shot alongside LIBERO-10 evaluation results presents a complete picture of DEPS’ capabilities - highlighting how DEPS can not only handle long-horizon skill chaining but also novel task generalization, all under the same general framework.
>
> We thank you for your continued interest and engagement, and hope we were able to thoroughly answer your questions.

---

### Official Review · Reviewer_pzRg · 2025-06-26

**Clarity:** 2
**Significance:** 2
**Originality:** 1
**Rating:** 4
**Confidence:** 3

**Summary:**

This paper presents an algorithm for the Discovery of GEneralizable Parameterized Skills (Deps). The proposed method learns skills from demonstrations in an end-to-end way, by training a 3-level architecture with a discrete skill selector, a continuous parameter selector conditioned on the discrete skill, and a subpolicy conditioned on both. The proposed method is evaluated on two multitask environments: LIBERO and MetaWorld, and compared against BC untrained and pretrained, and Prise.

**Questions:**

As mentioned, it seems that in some cases the additional complexity of the proposed method doesn't pay off in terms of improved performance, especially in longer horizon tasks. Can you point out which are the case where instead there is a clear benefit and the improved performance clearly justify the additional complexity?

Are discrete skills transferable from tasks to tasks?

Are discrete skills interpretable (eg grasp, lift, ...)?

What are low-level actions a_t (joint positions, velocities, cartesian, ...)? Is one choice of the other preferable in your formulation?

How do the representation of skills learned by your method compare to those from Prise's action abstractions?

[Figure 7] "We find that general skills corresponding to grasping, moving, and releasing objects are learned and consistently applied across visually diverse tasks." -- are these skills learned every time or are they "transferred" through the learned intermediate level of the architecture?

**Ethical Concerns:**

["NO or VERY MINOR ethics concerns only"]

**Final Justification:**

I believe my questions have been addressed, I confirm my score, and I have completed the mandatory acknowledgement.

**Limitations:**

Limitations are only discussed in the appendix. The limitations of performance on long horizon tasks is not actually addressed, but authors seem to hint at the fact that adopting a higher dimensional state for compression might help this issue. Are there any additional experiments pointing towards this intuition?

**Paper Formatting Concerns:**

No.

**Quality:**

3

**Strengths And Weaknesses:**

The paper is presented in a clear and well structured way. It nicely motivates the problem at hand and scopes it within relevant literature. The problem tackled is important and can have impact across various areas where decision making is used.
The method and it's mathematical formulation are presented clearly and looks correct to the best of my understanding.
The proposed algorithm is also evaluated and analyzed though several experiments, and compared to other methods. This helps understanding the method strengths in comparison to other approaches.
However, in my opinion, the originality of the method introduced is marginal, and the choice of a one-dimensional state compression is arguable: while the paper initially mentions manifolds as a geometry to describe this step, a one-dimensional state seems very limitative. And as authors mention in the limitation section in appendix, this in fact hinders performance in longer horizon tasks. This make me doubt the actual value of adopting the additional complexity of the method if gains are only observed in relatively short tasks.

---

> ### Author Rebuttal · Authors · 2025-07-31
>
> Thank you for the positive evaluation and thorough comments. We seek to address your questions below.
>
> **Q1: [paraphrased]**  The choice of a one-dimensional state compression is arguable
>
> **A1:** This is a natural question. To answer this question, we ran additional experiments to compare the mean success rates and the mean highest success rates on the LIBERO-OOD test setting for DEPS with 1D state compression (results from the paper) against DEPS with 3D state compression. The results for 3D state compression are averaged across 3 seeds.
>
> | Compression Dimension | Mean Success Rate | Mean Highest Success Rate |
> |----------------------|-------------------|---------------------------|
> | 1D (from paper)      | 0.34 ± 0.08      | 0.66 ± 0.12              |
> | 3D                   | 0.05 ± 0.04      | 0.17 ± 0.11              |
>
> The dramatic performance drop from 1D to 3D compression (0.34 → 0.05 mean success rate) suggests that aggressive compression to a single dimension is indeed crucial for DEPS effectiveness, likely due to reduced overfitting to training environments.
>
> In response to the reviewer’s comment that “authors seem to hint at the fact that adopting a higher dimensional state for compression might help…”, we would like to clarify this potential misunderstanding.  As can be seen in the empirical result above, in the case of LIBERO-OOD, adopting a higher dimension for state compression does not increase performance. In the Limitations Section, we suggest that training a residual policy [1] which has access to the full observation space and augments the output of the original subpolicy (which will still utilise 1D state compression), may be a promising approach - we leave an evaluation of this to future work.
>
> [1] Silver et. al, Residual Policy Learning. 2018
>
> **Q2:** As mentioned, it seems that in some cases, the additional complexity of the proposed method doesn't pay off in terms of improved performance, especially in longer horizon tasks. Can you point out which are the cases where, instead, there is a clear benefit and the improved performance clearly justifies the additional complexity?
>
> **A2:** You raise an important point about the relatively weaker performance of DEPS on  LIBERO-10. We would like to stress that the tasks in LIBERO-10 mostly consist of concatenations of tasks in the pretraining dataset, and therefore, there is a *minimal distribution shift between the pretraining and finetuning tasks in this setting.*  This is precisely the scenario where we'd expect less benefit from parameterized skills, as there is no need to generalize to new tasks or environments. We would like to note that despite the in-distribution nature of LIBERO-10, DEPS still performs better than PRISE and BC.
>
> We expect DEPS to perform strongly on long-horizon and *out-of-distribution* tasks, and leave an evaluation of this to future work.
>
> We believe that there are several scenarios where there is a clear benefit to using DEPS:
>
> 1. **Out-of-Distribution Generalization (LIBERO-OOD, page 7 Table 1):** DEPS shows substantial improvements when generalizing to entirely unseen environments and objects:
>
>  - Mean success: 0.34 ± 0.08 vs BC: 0.15 ± 0.04 (127% improvement)
>  - Mean highest success: 0.66 ± 0.12 vs BC: 0.36 ± 0.08 (83% improvement)
>
>    This represents the core scenario DEPS is designed for - adapting learned behavioral primitives to novel contexts. We believe this scenario is highly relevant for robotics applications, where robots have to deal with limited training datasets and extremely diverse tasks and environments..
>
> 2. **Low-Data Regimes (LIBERO-3-shot, Table 1):** For robotics applications, there often is very little demonstration data available for downstream tasks. When reducing the amount of training data available for downstream tasks, the gap between DEPS and other baselines increases.
>
>  - Mean success: 0.26 ± 0.03 vs BC: 0.11 ± 0.05 (136% improvement)
>  - Mean highest success: 0.49 ± 0.03 vs BC: 0.22 ± 0.08 (123% improvement)
>
> 3. **Limited Pretraining Compute (Table 2):** When reducing the amount of pretraining performed, DEPS significantly outperforms other baselines
>
>  - At 5 epochs: DEPS 0.24 ± 0.08 vs BC 0.07 ± 0.03 (243% improvement)
>  - At 15 epochs: DEPS 0.39 ± 0.05 vs BC 0.12 ± 0.06 (225% improvement)
>
>    This suggests that, in addition to improving downstream generalization to out-of-distribution tasks, DEPS’ learned abstractions improve the efficiency of pretraining (page 8, section 5.4).
>
>  We will add a discussion on the specific regimes where DEPS is useful in the revised version of the paper.
>
> **Q3:** Are discrete skills transferable from tasks to tasks?
>
> **A3:** Yes! As shown in Figure 7 (page 9), DEPS discovers semantically meaningful skills corresponding to action primitives such as grasping objects, closing doors etc. Importantly, we find that the discovered skills are applied *consistently* across tasks. An example can be seen in Figure 7 where the same grasp_object skill is used to pick butter in a cabinet environment  (top) and pick a mug (bottom) in a kitchen environment (bottom). This is not a cherry-picked example; we can release an annotated version of the entire LIBERO dataset across multiple tasks, showing that learned discrete skills consistently hold across distinct tasks and environments. The transferability of the learned discrete skills is supported by the observation that there is significant overlap in the distribution of continuous parameters used for a given discrete skill across different tasks. It is precisely this non-overfitted continuous parameterization that enables the discrete skills to easily generalize and transfer to unseen tasks.
>
> **Q4:** Are discrete skills interpretable (eg grasp, lift, ...)?
>
> **A4:** The learned skills are interpretable and correspond to action primitives such as object grasping/releasing, door closing, knob turning, etc. An example of this is provided in Figure 7. Furthermore, the learned continuous parameters represent smooth modulations in the behaviour of the discrete skill (visualizations are on our website, which was already included at the time of submission; the URL is in section 5.5).
>
> **Q5:** What are low-level actions $a_t$ (joint positions, velocities, cartesian, ...)? Is one choice of the other preferable in your formulation?
>
> **A5:** The action space in LIBERO consists of a 6-dimensional continuous-values vector representing OSC_POSE, which consists of the desired position and orientation of the end effector. In addition, there is a binary value representing the gripper state (opened/closed). In line with previous work, we train our policies to output a Mixture of Gaussians representing a desired distribution over actions.
>
> In Metaworld, the action space consists of a 3D vector representing the desired *change* in the end effector position as well as a binary variable representing the gripper (opened/closed). In line with previous work, we use a deterministic policy that outputs the desired action.
>
> We believe that DEPS is compatible with diverse low-level action types. Our empirical results on LIBERO and Metaworld use their standard respective environment action spaces, as also used in previous works, with minimal differences in the hyperparameters used.
>
> **Q6:** How do the representation of skills learned by your method compare to those from Prise's action abstractions?
>
> **A6:** Unfortunately, to the best of our knowledge, PRISE does not provide visualizations of its learned action abstractions.
>
> PRISE learns discrete action abstractions, each of which represents a fixed sequence of “action tokens”. We believe the parameterized skills learned by DEPS, which can be varied smoothly by changing the continuous parameter and do not have a fixed length (unlike in PRISE), present a more flexible and interpretable representation.
>
> **Q7:** [Figure 7] "We find that general skills corresponding to grasping, moving, and releasing objects are learned and consistently applied across visually diverse tasks." -- are these skills learned every time or are they "transferred" through the learned intermediate level of the architecture?
>
> **A7:** The learned skills are transferred across the intermediate level of the architecture. This is supported by several aspects of our experiments, including:
>
> 1. The same discrete skill is consistently applied in all relevant contexts. For example, discrete skill 9 (“grasp_object”) (as shown in Figure 7) is applied every time an object needs to be grasped, irrespective of the task.
>
> 2. Table 2 (page 8) shows that the gap between DEPS and other baselines increases when the number of pretraining epochs is decreased. This supports the argument that the learned skills are being used across tasks, therefore increasing the efficiency of pretraining through cross-task generalization.
>
> Thank you again for your constructive feedback, which has helped improve the clarity of several key aspects of our paper. Please let us know if you have any additional questions.

---

> > ### Comment · Reviewer_pzRg · 2025-08-06
> >
> > Thank you for the detailed rebuttal. The authors have adequately addressed my concerns, and I appreciate the clarifications provided.

---

### Official Review · Reviewer_7pDZ · 2025-07-01

**Clarity:** 3
**Significance:** 3
**Originality:** 3
**Rating:** 5
**Confidence:** 4

**Summary:**

The authors introduce an approach called deps which learns a skill parameterized policy from demonstrations.  The approach learns a discrete latent that encodes a set of skills and a continuous latent that modulates the skill. A key insight is compressing the state into a 1d vector to select along the manifold of skills. This forces the skill latents to encode useful information. The authors compare their approach to prior work and show that it outperforms these baselines in terms of task success rate.

**Questions:**

Please include an ablation analysis showing the benefit of compression and tradeoffs with different compression sizes for st. I will increase my score if this is provided.

Please explain if and why zt changes at each timestep

**Ethical Concerns:**

["NO or VERY MINOR ethics concerns only"]

**Final Justification:**

I appreciate the authors response and I think this paper should be accepted at Neurips. Therefore, i will keep my score of 5

**Quality:**

3

**Strengths And Weaknesses:**

Strengths - the is paper presents a sound and interesting approach that enables semantically skill learning. The authors provide convincing results that are both qualitative and quantitative. Their idea to compress observation embedding into a 1d vector to ensure that the skill space is forced to capture useful skill-related information. The separation of skill parameterization into a discrete and continuous latent make the learned policy more interpretable and successful in terms of generalization.

Fig 2. is confusing. I am not sure what the gray areas represent and the components could be explained better.

The paper could benefit from an ablation analysis showing that a 1d scalar is the best choice (why not 2d)?

A highly compressed st means that kt and zt are forced to encode all task-critical information - perhaps even information that isnt relevant to skill. Could the authors provide more compelling results showing that kt and zt encode only skill relevant information?

It is confusing that zt is sampled per kt but can be redefined at each timestamp. it would be helpful if the authors could further elaborate on what this means. If it can adapt at each timestamp, are there any constraints imposed on how it can adapt?

---

> ### Author Rebuttal · Authors · 2025-07-31
>
> We thank you for your thorough review and positive comments. We address your concerns below.
>
> **Q1:** Fig. 2 is confusing. I am not sure what the gray areas represent, and the components could be explained better.
>
> **A1:** As denoted in Figure 2’s legend, the gray area represents the variables that are observable to each policy/encoder. For example, the variational encoder can see the whole trajectory history and future, while the low-level policy can only see the previous transitions and the skills being selected at the current time step. We will clarify this further in the revision.
>
> **Q2:** Please include an ablation analysis showing the benefit of compression and tradeoffs with different compression sizes for $s_t$.
>
> **A2:** We thank the reviewer for the helpful suggestion. We have added a set of new experiments, comparing the mean success rates and the mean highest success rates on the LIBERO-OOD test setting for DEPS with 1D state compression (results from the paper) against DEPS with 3D state compression. The results for 3D state compression are averaged across 3 seeds.
>
> | Compression Dimension | Mean Success Rate | Mean Highest Success Rate |
> |----------------------|-------------------|---------------------------|
> | 1D (from paper)      | 0.34 ± 0.08      | 0.66 ± 0.12              |
> | 3D                   | 0.05 ± 0.04      | 0.17 ± 0.11              |
>
> The dramatic performance drop from 1D to 3D compression (0.34 → 0.05 mean success rate) suggests that aggressive compression to a single dimension is crucial for the method's effectiveness.
>
> **Q3:** A highly compressed st means that $k_t$ and $z_t$ are forced to encode all task-critical information - perhaps even information that isn’t relevant to skill. Could the authors provide more compelling results showing that $k_t$ and $z_t$ encode only skill-relevant information?
>
> **A3:** We thank the reviewer for the insightful comment; it is indeed important to ensure that $k_t$ and $z_t$ encode skill-relevant information rather than just general task information. One of DEPS’ main goals is to ensure this precise behavior. As outlined in Appendix A1, during pretraining, the variational network predicts continuous parameters per discrete skill for each forward pass over a given trajectory (instead of predicting a new continuous parameterization at each time step, which could lead to overfitting).  Additionally, to prevent the continuous parameters from overfitting to specific tasks, we introduce the $\mathcal{L}_{\text{norm}}$ (Appendix A1) regularization term, which bounds the range of the learned continuous parameters. This limits the amount of information encoded in the discrete skill and continuous parameters, forcing the 1D state to encode meaningful information as an “index” into parameterized skill trajectories.  This is validated in Figure 8, where the learned latent 1D states vary monotonically within discrete skills, with discontinuities when the discrete skill switches. The strong performance of DEPS on out-of-distribution downstream tasks (Table 1, LIBERO-OOD) further supports our hypothesis that DEPS learns meaningful abstractions instead of overfitting to $k$ or $z$ values. This is particularly evident in low-data regimes (Table 1, LIBERO-3-shot) and low pretraining settings (Table 2).
>
> We provide several lines of evidence that $k_t$ and $z_t$ do encode skill-relevant information:
>
> 1. **Consistent application of discrete skills (Section 5.5 and Figure 7):** We find that DEPS discovers semantically meaningful discrete skills corresponding to primitive behaviors such as grasping objects, opening doors, etc. Importantly, we find that the discovered skills are applied *consistently* across tasks. An example can be seen in Figure 7, where the same ``grasp_object’’ skill is used to pick butter in a cabinet environment (top) and pick a mug (bottom) in a kitchen environment (bottom). This is not a cherry-picked example; we can release an annotated version of the entire LIBERO dataset across multiple tasks, showing that learned discrete skills consistently hold across distinct tasks and environments.
>
> 2. **Smooth variation in policy on varying continuous parameters (Section 5.5):** We find that for a given discrete skill, slight modifications in the continuous parameter result in smooth variations in the resulting policy. For example, modifying the continuous parameters for a discrete skill used to turn a knob smoothly varies the location at which the knob turning action is executed. Visualizations of this are provided on our website (which was already included at the time of submission; the URL is in Section 5.5).
>
> 3. **Overlap of skill-specific continuous parameterizations across tasks:** We have visualizations that show that for a given discrete skill, there is significant overlap in the continuous parameters chosen for different tasks, which suggests that $z$ only encodes skill-specific, and not task-specific, information. We are happy to update the project website with these visualizations for the reviewers' interest, provided the AC gives approval or clarification on the website update policy; regardless, we will include such visualizations in the final camera-ready.
>
> **Q4:** Please explain if and why $z_t$ changes at each timestep.
>
> **A4:** *During pretraining*, $z_t$ is not changed at every timestep. Instead, $z_t$ can only change when the sampled discrete skill changes. This is essential to ensure that the learned parameterized skills are temporally extended, and helps prevent the model from taking degenerate shortcuts (such as encoding the action at each timestep directly in the continuous parameter) that would prevent the model from learning meaningful abstractions. Details on this can be found in Appendix A1.
>
> On the other hand, *during finetuning*, we allow the model to choose to change its choice of continuous parameter at each timestep if necessary. Our goal during the finetuning phase is to tune the abstractions learned during pretraining to maximize performance on downstream tasks. By modifying its continuous parameter at each timestep, the robot can “close the planning loop” and therefore more flexibly react to potential stochasticity (or unexpected outcomes) at inference time. Details on this can be found in Appendix A2.
>
> In summary, during pretraining, our goal is to learn reusable abstractions/compressions, and therefore, we do not change $z_t$ at each timestep to encourage compression. On the other hand, during inference, we allow $z_t$ to change at each timestep as our goal is to use learned specialized abstractions to maximize performance on a task.
>
> We appreciate the reviewer’s thoughtful comments and suggestions on strengthening our work, and welcome further opportunities to clarify any additional questions that may remain.

---

> ### Comment · Reviewer_7pDZ · 2025-08-06
> **Response to authors**
>
> Thank you for the clarifications. I find the result of decreased performance with 3D compression very interesting. I think it would add to the paper if the authors include this result and short discussion about why they see this result. Otherwise, the authors have adequately addressed my concerns and I will keep my score of 5

---

> > ### Author Response · Authors · 2025-08-07
> > **Additional State Compression Ablation Results**
> >
> > We are happy to have addressed your concerns and appreciate your interest in the state compression ablation. We would like to bring to attention some additional results we have on this ablation - they are available in our general comment, but also pasted below for the reviewer’s convenience:
> >
> > | Compression Dimension | Mean Success Rate | Mean Highest Success Rate |
> > |----------------------|-------------------|---------------------------|
> > | 1D (from paper)      | 0.34 ± 0.08      | 0.66 ± 0.12              |
> > | 2D                   | 0.13 ± 0.12      | 0.30 ± 0.18              |
> > | 3D                   | 0.05 ± 0.04      | 0.17 ± 0.11              |
> >
> >
> > As can be seen in the presented table, there is a **clear monotonic decline** in performance as the state compression dimension is increased from 1D to 3D.
> >
> > In Section 4.2, we explain how state compression is essential to learn meaningful abstractions and generalize to unseen environments/tasks. A natural question that follows is whether aggressive compression to 1D is overly restrictive. As we explain in the section, we believe *this is not the case*, as a parameterized skill can be viewed as a *family of related parameterized trajectories.* Each trajectory is defined by a choice of continuous parameterization, and given a fixed trajectory, a 1D state is sufficient as one only needs an “index” into the trajectory (Figure 3). As the dimensionality of the state embedding increases, we expect more information to be encoded in the state embedding, which implies (i) lower overlap in the embedded state-space across tasks and (ii) less reliance on the latent variables ($k$ and $z$) to encode meaningful information, resulting in poorer performance in unseen tasks/environments. This intuition matches the empirical results of monotonically decreasing performance shown in the table above.
> >
> > We agree with the reviewer that these ablation results are interesting and insightful, and we will certainly add them as well as their relevant discussions in the final version of the paper.

---

### Author Response · Authors · 2025-08-07
**Additional Results on State Compression Ablation**

We would like to thank all reviewers for their comments so far. A common thread amongst reviewers has been the need for an ablation showing that DEPS’ choice of 1D state compression is central to its performance. While we included a comparison between 1D and 3D state compression in our original responses, we have since also evaluated DEPS with 2D state compression and present the updated results below (the results on 2D and 3D compression are averaged over three seeds):

| Compression Dimension | Mean Success Rate | Mean Highest Success Rate |
|----------------------|-------------------|---------------------------|
| 1D (from paper)      | 0.34 ± 0.08      | 0.66 ± 0.12              |
| 2D                   | 0.13 ± 0.12      | 0.30 ± 0.18              |
| 3D                   | 0.05 ± 0.04      | 0.17 ± 0.11              |

We see a **clear monotonic decrease** in performance as the dimensionality of state embeddings increases, validating our motivation to use 1D state “indices” as explained in Section 4.2.

---

### Decision · Program_Chairs · 2025-09-17

**Decision:**

Accept (poster)

**Comment:**

This paper introduces DEPS, an end-to-end algorithm for learning parameterized skills from expert demonstrations, which is a non-trivial and valuable contribution. Experiments on LIBERO and MetaWorld show improved generalization and performance over baseline methods.

The main concerns include (1) limited analysis of the 1D state compression technique, (2) reliance on non-standard evaluation metrics, (3) the use of relatively weak baselines for LIBERO and MetaWorld, and (4) modest performance partly due to weak base models. The paper could be strengthened by providing deeper interpretation and validation of the compression technique, reporting results with standard imitation learning metrics (e.g., success rates at convergence or fixed training steps), and demonstrating scalability with more advanced methods (e.g., action chunking) and stronger baselines (e.g., ACT, Diffusion Policy).

Overall, I lean toward accept, as the paper presents a non-trivial and well-executed approach to parameterized skill discovery from demonstrations, with potential for meaningful impact despite the noted limitations.